# Perivascular niche cells sense thrombocytopenia and activate hematopoietic stem cells in an IL-1 dependent manner

Tiago C. Luis [1,2,3,4] ✉, Nikolaos Barkas[1,2], Joana Carrelha[1,2], Alice Giustacchini [1,2,5,6], Stefania Mazzi[7], Ruggiero Norfo[1,2], Bishan Wu[1,2], Affaf Aliouat[1,2], Jose A. Guerrero [8,9], Alba Rodriguez-Meira[1,2], Tiphaine Bouriez-Jones[1,2], Iain C. Macaulay [1,2,10], Maria Jasztal[8,9], Guangheng Zhu[11,12], Heyu Ni [11,12,13,14], Matthew J. Robson [15,16], Randy D. Blakely[15], Adam J. Mead [1,2], Claus Nerlov [2], Cedric Ghevaert[8,9] & Sten Eirik W. Jacobsen [1,2,7,17,18] ✉

Hematopoietic stem cells (HSCs) residing in specialized niches in the bone marrow are responsible for the balanced output of multiple short-lived blood cell lineages in steady-state and in response to different challenges. However, feedback mechanisms by which HSCs, through their niches, sense acute losses of specific blood cell lineages remain to be established. While all HSCs replenish platelets, previous studies have shown that a large fraction of HSCs are molecularly primed for the megakaryocyte-platelet lineage and are rapidly recruited into proliferation upon platelet depletion. Platelets normally turn-over in an activation-dependent manner, herein mimicked by antibodies inducing platelet activation and depletion. Antibody-mediated platelet activation upregulates expression of Interleukin-1 (IL-1) in platelets, and in bone marrow extracellular fluid in vivo. Genetic experiments demonstrate that rather than IL-1 directly activating HSCs, activation of bone marrow Lepr⁺ perivascular niche cells expressing IL-1 receptor is critical for the optimal activation of quiescent HSCs upon platelet activation and depletion. These findings identify a feedback mechanism by which activation-induced depletion of a mature blood cell lineage leads to a niche-dependent activation of HSCs to reinstate its homeostasis.

Hematopoietic stem cells (HSCs) are responsible for sustaining the production of all blood cell lineages throughout life. Whereas in unperturbed adult hematopoiesis HSCs are mainly in a quiescent or slowly proliferative state, they can become rapidly activated in response to different types of stress including infection and cytotoxic treatments[1–3]. This is compatible with the existence of feedback mechanisms ensuring an on-demand replenishment of blood cell lineages. While previous studies have implied different mechanisms by which HSCs can respond to pro-inflammatory signals such as interferons[4,5], feedback mechanisms activating quiescent HSCs in a niche-dependent manner upon physiological stress, including in response to selective loss of a single blood cell lineage, remain to be identified[1–3].

Recent studies suggested that selective loss of granulocytes results in activation of bone marrow (BM) progenitors rather than HSCs[6] and that granulocyte-colony stimulating factor (G-CSF), a physiological regulator of granulocyte production, stimulates granulocyte progenitors but not HSCs[2]. In contrast, Thrombopoietin (THPO), the key cytokine regulator of megakaryopoiesis and platelets[7], also plays an important role in the maintenance of HSCs which express high levels of the THPO receptor MPL[8–10]. HSCs transcriptionally primed for the megakaryocyte-platelet lineage are activated in response to acute platelet depletion[9], but it is unclear whether THPO, which is produced in the distant liver and directly activates MPL-expressing HSCs[11], can fully account for the activation of HSCs in the BM[12–14]. In fact, *Mpl* knockout mice still produce platelets under stress situations[14], implicating THPO-independent mechanisms, potentially through supporting BM niche cells critical in regulating HSC function[15]. Other studies demonstrated that acute and chronic thrombocytopenia can activate HSCs, implicating specific regulators and accompanying changes in the BM microenvironment[16,17], although genetic experiments to specifically establish the role of these regulators in BM HSC niche cells were not performed.

Herein we set out to investigate feedback mechanisms responsible for activation of HSCs in response to acute platelet depletion. We hypothesized that distinct BM niche cells might be poised to sense peripheral platelet depletion and subsequently activate otherwise quiescent HSCs. We identified a mechanism by which IL-1, released upon platelet activation and subsequent depletion, is sensed by a distinct population of IL-1R+ perivascular niche cells in the central BM, which upon IL-1-activation stimulate HSCs to re-establish platelet homeostasis.

## Results

### Acute platelet depletion rapidly recruits quiescent HSCs into proliferation

In line with previous studies[9], administration of a single dose of anti-GPIbα antibody to *Vwf*-GFP mice, to mimic the thrombocytopenia observed in patients with immune thrombocytopenia purpura (ITP), rapidly and efficiently depleted platelets in the course of 1 day without significantly affecting other blood cell lineages (Fig. 1a). This was accompanied by a rapid (within 1 day) cell cycle activation of Lin−Sca1+c-Kit+ (LSK) Flt3−CD48−CD150+ HSCs. Recruitment into active cell cycle was preferentially observed within the *Vwf*-GFP+ (Vwf+) HSC compartment (Fig. 1b–d), which is enriched in platelet-biased HSCs but predominantly contains multilineage reconstituting HSCs[9,18]. While over 60% of Vwf+ LSKFlt3−CD48−CD150+ HSCs entered the S-G2-M phase of cell cycle, only a small fraction of *Vwf*-GFP− (Vwf−) HSCs entered S-G2-M. Nevertheless, most Vwf− HSCs moved from a quiescent (G0) to a more activated G1 stage (Fig. 1c, d).

We next sought to further explore the mechanistic basis for the HSCs activation observed after platelet depletion. The anti-GPIbα antibody binds to GPIbα (CD42b), the receptor for thrombin and VWF[19], leading to platelet activation, desialylation and subsequent clearance in an Fc-receptor independent manner[20]. GPIbα is specifically expressed in the megakaryocytic (Mk)-platelet lineage, including on MkPs, but importantly for our studies and as previously reported by others[21], GPIbα expression is virtually undetectable on HSCs (Supplementary Fig. 1a). Following the initial platelet depletion, platelet numbers slowly recovered, with normal platelet counts being re-established between day 5 and 10 post-depletion (Fig. 1a). Notably, this platelet depletion led to a 4-fold increase in the number of Vwf+ LSKFlt3−CD48−CD150+ HSCs by 2 days, while Vwf− HSCs increased with slower kinetics following an initial reduction (Fig. 1e). Subsequently, the Vwf+/Vwf− HSC ratio and absolute numbers gradually returned back to normal, concomitantly with the normalization of platelet counts (Fig. 1a, e). The increase in Vwf+ HSCs was also accompanied by a selective increase in the numbers of the LSK Flt3−CD48+CD150+

subset of MPPs (Fig. 1b and Supplementary Fig. 1b) shown to be Mk-biased[22]. Furthermore, MkPs[23] were also significantly increased a few days after the increase in Vwf+ HSCs, whereas erythroid (Pre-CFU-E) and myeloid (GMP) progenitors were not significantly affected (Supplementary Fig. 1c, d). Of note, despite their robust GPIbα expression and in agreement with the Fc-independent mechanism by which the antibody leads to platelet depletion[20], MkPs were initially slightly (although not significantly) reduced whereas Mks were not depleted in BM by antibody administration (Supplementary Fig. 1d–g). Following the rapid initial activation, HSCs quickly returned back to quiescence with a normalized cell cycle phase distribution being observed already 5 days post platelet depletion (Fig. 1c, d). An alternative monoclonal antibody (NIT E)[20] depleted platelets and activated Vwf+ and Vwf− HSC cell cycle in a similar manner as the anti-GPIbα antibody (Supplementary Fig. 2a, b). GPIbα antibody treatment was accompanied by a mild and transient splenomegaly with increased numbers of Vwf+ but not Vwf− HSCs in the spleen (Supplementary Fig. 2c, d).

The recruitment of quiescent LSKFlt3−CD48−CD150+ BM cells into proliferation upon acute platelet activation and depletion implicate the existence of a feedback mechanism by which quiescent HSCs are recruited to re-establish Mk and platelet homeostasis. In order to investigate if HSCs from platelet-depleted mice are more efficient in generating Mks, we used a single-cell Mk/GM in vitro differentiation assay. Clonogenicity of single Vwf+ or Vwf− LSKFlt3−CD48−CD150+ HSCs isolated from mice in homeostasis (IgG) or after platelet depletion (GPIbα) was similar (~80%; Supplementary Fig. 2e). However, single Vwf+ HSCs isolated from platelet-depleted mice differentiated faster into Mks than Vwf− HSCs from platelet-depleted mice or Vwf+ HSCs isolated from mice in homeostasis (Fig. 1f) and also generated a higher number of colonies exclusively consisting of Mks (Fig. 1g and Supplementary Fig. 2f, g). Stem-like Mk-committed progenitors phenotypically resembling HSCs have been previously shown to proliferate in response to poly(I:C)-induced thrombocytopenia[24]. Therefore, to more definitively establish that LSKFlt3−CD48−CD150+ BM cells induced to proliferate in response to anti-GPIbα induced platelet depletion include true long-term repopulating HSCs, we used doxycycline-inducible (tet-ON) H2B-mCherry mice[6,25]. In this system, a pulse of doxycycline treatment results in incorporation of mCherry-labeled histones into nucleosomes, which divide equally between daughter cells when cells proliferate. In agreement with the cell cycle analysis (Fig. 1c, d), anti-GPIbα induced platelet depletion resulted in increased proliferation of HSCs, as evidenced by the increased dilution of the mCherry labeling after 3 days (Supplementary Fig. 2h). Importantly, FACS sorted mCherry^lo (proliferative) LSKFlt3−CD48−CD150+ cells sorted from platelet-depleted mice had in vivo long-term (LT; 16 weeks) multilineage reconstitution potential (Supplementary Fig. 2i), demonstrating that anti-GPIbα induced platelet depletion recruits potent LT-HSCs into proliferation. To more directly compare Vwf+ HSCs that become activated or that remain in a non-proliferating state following platelet depletion we used a non-invasive labeling method based on the injection of the N-hydroxilsulfosuccinimide biotin derivative (Biotin), which efficiently labels the membrane proteins of all BM cells[26]. When cells divide, labeled membrane proteins are equally distributed among daughter cells allowing the analysis of cell division history in vivo. Similarly, to the H2B-mCherry model, platelet depletion resulted in reduced biotin-labeling of Vwf+ HSCs (Fig. 1h) and also of Vwf− HSCs (Supplementary Fig. 2j). The differences observed in the cell cycle and proliferation analyses of Vwf− HSCs might potentially reflect the previously described hierarchical relationship between Vwf+ and Vwf− HSCs[9]. Thus, proliferating Vwf+ HSCs may differentiate into Vwf− HCS, which carry over the history of biotin label dilution. Importantly, FACS sorted Vwf+Biotin^lo (proliferative) LSKFlt3−CD48−CD150+ cells sorted from platelet-depleted mice (Supplementary Fig. 2k, l) had in vivo long-term (LT; 16 weeks) multilineage

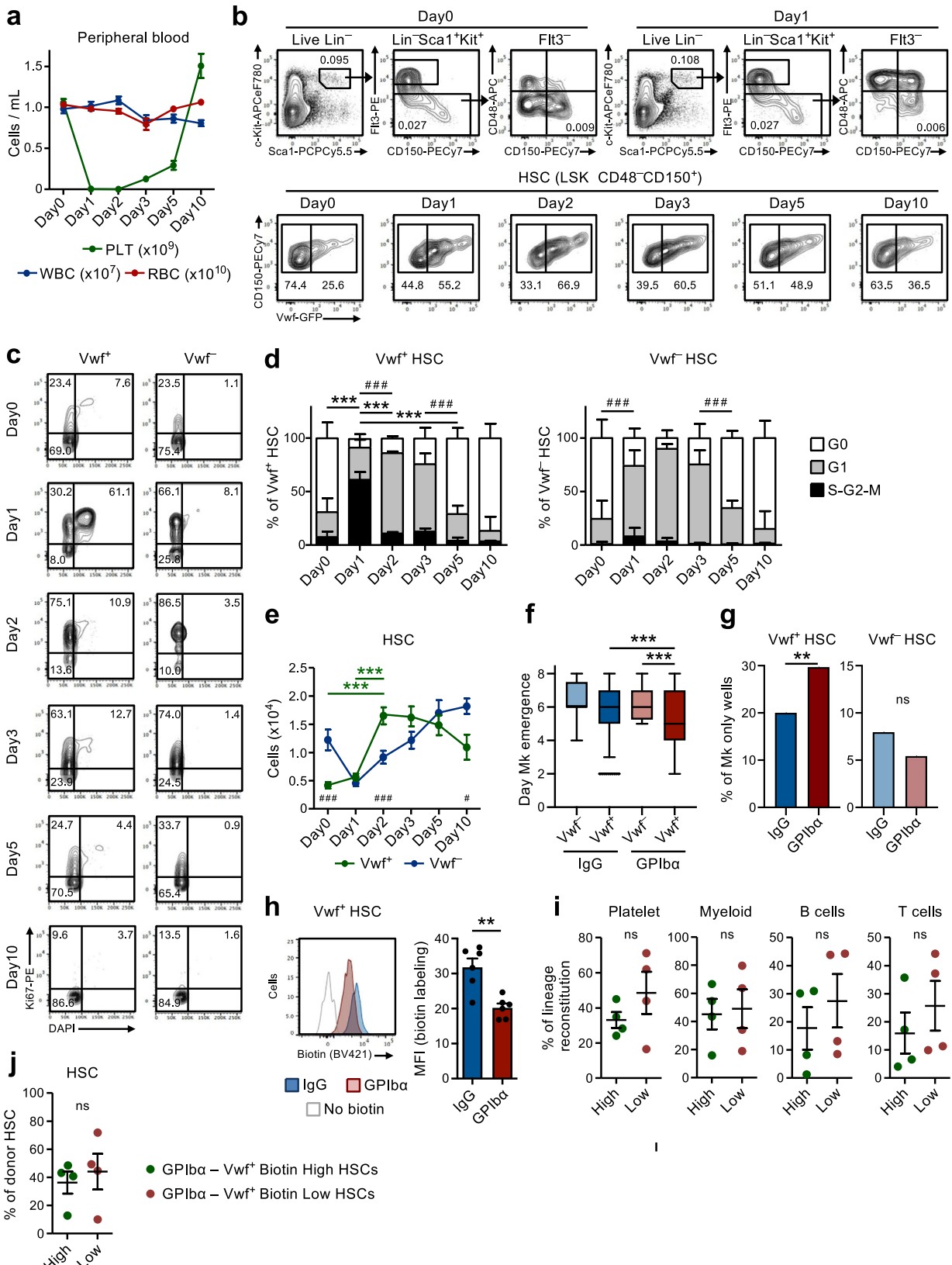

reconstitution potential (Fig. 1i), demonstrating that anti-GPIbα induced platelet depletion recruits potent LT-HSCs into proliferation. While potently replenishing platelets, no significant platelet bias was observed of the Vwf+ HSCs proliferating in response to platelet depletion, in comparison to non-proliferative (Biotin^Hi) Vwf+ HSCs (Fig. 1i). Regardless, both biotin fractions of Vwf+ HSCs demonstrated

similar LT-HSC reconstitution (Fig. 1j) and were overall equally efficient at generating the Vwf+ and Vwf− HSC compartments (Supplementary Fig. 2m). This is in line with functionally defined LT-HSCs being recruited into proliferation and with the fact that the relative proportions of Vwf+ and Vwf− HSCs largely return to normal once homeostasis has been re-established (Fig. 1b, e).

**Fig. 1 | Rapid activation of Vwf⁺ HSCs in response to acute platelet depletion precedes restoration of platelet homeostasis.** (Related to Supplementary Figs. 1 and 2). **a** Kinetics analysis of peripheral blood cell parameters post administration of anti-GPIbα antibody. Day 0 mice were treated with IgG isotype control antibody. Data represent mean ± SEM of 10 (Day0), 13 (Day1), 10 (Day2), 9 (Day3), 8 (Day5) and 7 (Day10) mice from 14 independent experiments. PLT platelets, WBC white blood cells, RBC red blood cells. **b** Representative FACS profiles and gating strategy of *Vwf*-GFP⁺ (Vwf⁺) and *Vwf*-GFP⁻ (Vwf⁻) LSKFlt3⁻CD150⁺CD48⁻ HSCs at the indicated time points after platelet depletion. Numbers in gates/quadrants indicate the frequency (average of all mice analyzed) of the gated cell population among total live cells (upper panels) or among HSCs (lower panels). **c, d** Cell cycle analysis of Vwf⁺ and Vwf⁻ LSKFlt3⁻CD150⁺CD48⁻ HSCs at the indicated time points post platelet depletion. **c** Representative cell cycle FACS profiles of Vwf⁺ (left) and Vwf⁻ (right) HSCs in G0 (DAPI⁻Ki67⁻) G1 (DAPI⁻Ki67⁺) or S-G2-M (DAPI⁺Ki67⁺) phases of cell cycle. Numbers in gates represent frequencies (average of all mice analyzed) of total HSCs. **d** Mean ± SD cell cycle phase distribution of Vwf⁺ (left) and Vwf⁻ (right) HSCs. Data from 5 (Day0), 5 (Day1), 3 (Day2), 5 (Day3), 5 (Day5) and 4 (Day10) mice from 6 independent experiments. ***$p < 0.001$; **$p < 0.01$ for the S-G2-M cell cycle fraction; ###$p < 0.001$ for the G1 cell cycle fraction (both using 2-way ANOVA with Tukey's multiple comparisons); **e** Absolute numbers of Vwf⁺ and Vwf⁻ HSCs (per 2 legs, see methods). Mean ± SEM data of 8 (Day0), 8 (Day1), 7 (Day2), 5 (Day3), 6 (Day5) and 5 (Day10) mice from 9 independent experiments. ***$p < 0.001$ for Vwf⁺

HSC (2-way ANOVA with Tukey's multiple comparisons); #$p < 0.05$ and ###$p < 0.001$ for the comparison of Vwf⁺ vs. Vwf⁻ HSCs (2-way ANOVA with Sidak's multiple comparisons). Time of appearance of the first Mk (**f**) and frequency of colonies with only Mk cells (**g**) in cultured single Vwf⁻ or Vwf⁺ HSCs isolated from mice 16 hrs post IgG or GPIbα treatment. Data from 138, 364, 147 and 451 single cell-derived colonies analyzed, respectively, from 5 biological replicates in 4 independent experiments. **f** Middle line represents median, box limits represent the 25–75 percentiles, whiskers mark the 5–95 percentiles. Cells outside the 5–95 percentiles are marked as outliers. *P* values calculated with Kruskal–Wallis test with Dunn's multiple comparisons. **g** *P* value calculated with two-sided Fisher's exact test. ***$p < 0.001$; **$p < 0.01$; *$p < 0.05$; ns, non-significant ($p > 0.05$). **h** Biotin proliferation analysis of *Vwf*-GFP⁺ HSCs 2 days post IgG or GPIbα treatment. Representative plot (left) and mean ± SD MFI (normalized for MFI of No biotin labeling control; right) from 6 mice per group in 3 independent experiments. **$p < 0.01$; calculated with two-sided t-test. Long-term reconstitution (16 weeks) of platelet, myeloid and lymphoid cell lineages in blood (**i**) and of the BM HSC compartment (**j**) by biotin high and biotin low *Vwf*-GFP⁺ HSC fractions 2 days post platelet depletion. 50 cells transplanted per mouse. Data represent mean ± SEM of 4 donors in 2 independent experiments. Each dot represents the mean of 2 recipient mice transplanted per donor. ns, non-significant ($p > 0.05$); calculated with two-sided t-tests. See also Supplementary Figs. 1 and 2.

## Transcriptional reprogramming of the HSC niche in response to acute platelet depletion

The existence of a feedback mechanism by which quiescent HSCs are recruited into proliferation in response to platelet depletion implies that HSCs in the BM must be able to sense the demand for platelet production. We hypothesized that stromal niche cells in the bone marrow might be involved in this process. In fact, changes in the BM niche after platelet depletion have been previously suggested[16,17] but the involvement of specific niche cells in regulating HSC function in this context has not been directly demonstrated. Therefore, we used global RNA-sequencing analysis of previously characterized BM niche cell populations as well as HSCs to investigate possible molecular interactions between these cells, involved in the HSC response to platelet depletion. For this, non-hematopoietic cells were isolated from two described[27–29] distinct anatomical regions in BM: the central BM (CBM) cells and the bone lining (BL) cells (Supplementary Fig. 3a). Within the CD45⁻Ter119⁻ non-hematopoietic cells in both BL and CBM fractions, distinct niche cell populations are defined as CD31^Hi endothelial cells (EC) and CD31⁻Lepr⁺ perivascular cells (PV). In the BL fraction we further defined CD31⁻Alcam⁻Pdgfrα⁺Sca1⁺ (PαS) mesenchymal progenitors, as well as CD31⁻Alcam⁻Sca1⁻*Osx*-GFP⁺ osteoblast progenitors (OBP) and CD31⁻Alcam⁺Sca1⁻ osteoblasts (OB) (Fig. 2a, b). Flow cytometric analysis of these cells isolated from mice in homeostasis and 1 day after platelet depletion did not reveal major changes in cellular composition, besides a small but significant increase in CBM-PV cells (Fig. 2a, b). RNA-sequencing analysis showed distinct clustering of the different endothelial, mesenchymal and HSC populations (Supplementary Fig. 3b). The expression of known markers defining the distinct niche cell populations (Fig. 2c and Supplementary Data 1) and, of different hematopoietic regulators clustered by pattern of gene expression (Supplementary Fig. 3c and Supplementary Data 1), further confirmed their distinct cell identities. Additionally, principal component analysis (PCA) of the niche cell populations showed the separation of the endothelial and mesenchymal lineage cells along principal component (PC)1 axis and, a further separation of mesenchymal populations along PC2 axis based on stage of differentiation (Fig. 2d). Differential gene expression analysis, comparing niche cells isolated from mice in homeostasis and 1 day post platelet depletion, showed a high number of differentially expressed (DE) genes in CBM-EC (266 genes) and CBM-PV (249 genes) cells (Fig. 2e, f and Supplementary Data 2, 3). In contrast, a much lower number of DE genes were detected in other niche populations, including the corresponding bone-associated BL-EC and BL-PV) (Fig. 2e, f, Supplementary Fig. 3d, e and

Supplementary Data 2–4), suggesting a preferential CBM niche involvement in the response to thrombocytopenia. Gene ontology (GO) analysis performed on the DE genes from CBM-EC highlighted biological processes associated with platelet activation/coagulation (*Pf4*, *Clec1b*, *P2ry12*), response to stress (*Ifitm3*, *Litaf*, *S100a4*) and cell adhesion (*Vcam1*, *Ctgf*) (Fig. 2g and Supplementary Data 5). GO analysis on CBM-PV DE genes revealed biological processes associated with inflammation and in particular with the cellular response to the pro-inflammatory Interleukin-1 (IL-1) (Fig. 2g and Supplementary Data 5). Other biological processes highlighted in CBM-PV cells were Toll-like receptor (TLR)-4 signaling, for which signaling pathways downstream of the receptor are partially shared with IL-1 signaling, and also mesenchymal differentiation and TGFβ signaling (*Vasn*, *Fst*, *Wisp2*, *Cyr61*) (Fig. 2f, g).

## IL-1 signaling in non-hematopoietic cells is critical for the HSC response to platelet depletion

Gene set enrichment analysis (GSEA) confirmed the enrichment of genes linked with inflammatory response and IL-1 receptor signaling observed in CBM-PV cells isolated from GPIbα-treated mice (Fig. 3a). These genes included *Il1rn* (IL-1 target gene and IL-1 signaling pathway antagonist), *Socs3* and *Dusp1* (regulators of the IL-1 signaling pathway) and *Fosb*, *Junb* and *Nfkbia* (components of pathways activated downstream of the IL-1 signaling cascade) (Fig. 3b). To further investigate a role for sterile inflammation in this process we analyzed the levels of different pro-inflammatory cytokines in the BM extracellular fluid of GPIbα-treated mice. This analysis revealed significantly increased levels of both IL-1α and IL-1β post platelet depletion (Fig. 3c), with kinetics paralleling the expansion and subsequent normalization of Vwf⁺ HSC (Fig. 1e). A similar initial increase was also seen for TNFα although this was sustained beyond the time at which Vwf⁺ HSC cell cycle status had normalized (Supplementary Fig. 4a). Other inflammatory cytokines were largely unchanged (IL-6 and IL-12) or decreased at later stages (INFγ) post platelet depletion (Supplementary Fig. 4a). In steady-state, IL-1 has been shown to mainly be produced by circulating T cells (IL-1α) and granulocytes (IL-1β)[30]. In our datasets analyzing for IL-1 expression both *Il1a* and *Il1b* were mostly undetectable in the different niche cell populations as well as in HSCs (Supplementary Fig. 4b), whereas primary Mks (Supplementary Fig. 4c, d) showed high expression, in particular of *Il1a* (Supplementary Fig. 4b).

Administration of recombinant IL-1 to mice resulted in the cell cycle activation of Vwf⁺ and Vwf⁻ HSCs (Fig. 3d), with Vwf⁺ HSCs being recruited at a higher extent into the S-G2-M phase of cell cycle

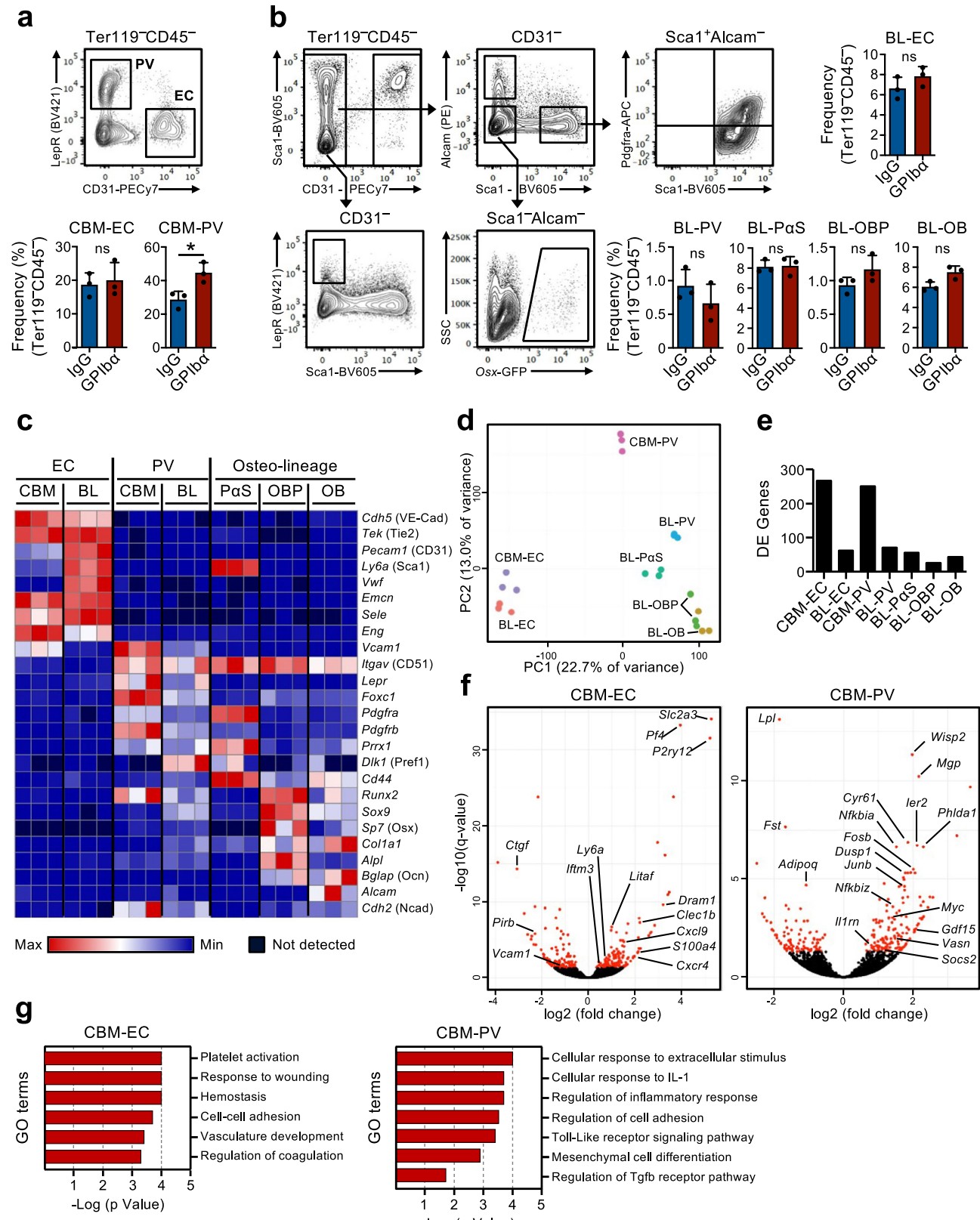

(Supplementary Fig. 4e), compatible with IL-1 mediating the HSC activation in response to anti-GPIbα-induced thrombocytopenia. To more specifically investigate this hypothesis, we induced platelet depletion in IL-1R deficient mice (*Il1r1⁻/⁻*), which have a normal HSC compartment[30] and normal platelet numbers. Importantly, *Il1r1⁻/⁻* mice showed a significant reduction in the frequency of actively cycling (S-

G2-M) Vwf⁺ HSCs after platelet depletion (Fig. 3e, f), in comparison to GPIbα-treated wildtype (Wt) mice. The fact that *Il1r1* deficiency did not completely abrogate cell cycle activation of HSCs in response to GPIbα-treatment suggests that other (IL-1 independent) mechanisms are involved. In agreement with this we previously showed increased levels of THPO in serum 1 day post platelet depletion[9]. To identify

**Fig. 2 | Up-regulation of a pro-inflammatory gene signature in bone marrow niche cells after acute platelet depletion.** (Related to Supplementary Fig. 3). FACS analysis and gating strategies for sorting of endothelial and stromal cells in the central bone marrow (CBM; **a**) and bone lining (BL; **b**) cell compartments of mice 1 day post platelet depletion (GPIbα antibody treatment). Control mice received isotype (IgG) control antibody. Bar diagrams represent mean ± SD frequencies (%) of each cell population among total non-hematopoietic CD45⁻Ter119⁻ cells. Data are from 3 mice per group in 3 (**a**) and 2 (**b**) independent experiments. *$p < 0.05$; ns non-significant ($p > 0.05$); assessed by two-sided t-test. **c–g** RNA-sequencing analysis of the endothelial/stromal cell compartments of mice 1 day post platelet depletion. **c** Expression (FPKM) of genes characterizing the different niche cell populations. **d** Principal component analysis of normalized gene expression of the different cell populations investigated. **e** Number of differentially expressed (DE) genes between IgG and GPIbα treated mice (adjusted *p* value (*q*)<0.05), in each niche cell population investigated. **f** Volcano plots and **g** gene ontology (GO) terms analysis of genes differentially expressed in CBM endothelial cells (EC) and Lepr⁺ perivascular (PV) cells. In **f**, red dots indicate significantly DE genes (*q* < 0.05. For all panels data represent mean ± SD FPKM of 3 biological replicates from 2 independent experiments. OB osteoblasts, OBP osteoblast progenitors, PαS Pdgfrα⁺Sca1⁺ mesenchymal progenitors. See also Supplementary Fig. 3.

other signals potentially synergizing with IL-1 in the activation of HSCs post platelet depletion we analyzed the expression of known HSC regulators, including *Tgfb1* and *Pf4*, previously implicated in HSC quiescence[31,32]; and *Fgf1*, previously associated with HSC proliferation[32]. RNA-sequencing analysis of distinct niche cell populations revealed a > 20-fold up-regulation of *Pf4* in CBM-EC (Supplementary Fig. 4f), while no change in the expression of *Tgfb1* and *Fgf1* was observed. We further investigated the protein levels of these regulators in the BM extracellular fluid post platelet depletion. In line with the gene expression analysis, TGFβ1 and FGF1 levels were not altered, but PF4 was significantly increased 1 day post GPIbα-mediated platelet depletion (Supplementary Fig. 4g). Given the previously described role of PF4 in inducing HSC quiescence[31,33], the observed increased levels of PF4 in BM extracellular fluid are unlikely to explain the activation of HSC proliferation post platelet depletion.

RNA-sequencing revealed very low levels (≈1 FPKM) of *Il1r1* transcripts in HSCs (Fig. 3g). Flow cytometric analysis showed undetectable levels of IL-1R protein on HSCs and low levels on several MPP subsets (Fig. 3h and Supplementary Fig. 5a). In addition to the lack of detectable IL-1R expression, none of the IL-1 signaling associated genes upregulated in CBM-PV were found to be upregulated in Vwf⁺ or Vwf⁻ HSC post GPIbα-treatment (Supplementary Fig. 5b). While we did not observe detectable IL-1R expression in phenotypically defined HSCs, IL-1 has been previously suggested to activate HSCs[30]. Therefore, to investigate if IL-1 may directly mediate the activation of HSCs post anti-GPIbα-induced thrombocytopenia we induced platelet depletion in *Il1r1*^FL/FL^ *Vav*-iCre^Tg/+^ mice which targets deletion of *Il1r1* to all hematopoietic cells, including HSCs. Droplet digital PCR analysis of Vwf⁺ and Vwf⁻ HSCs confirmed >99% deletion efficiency of the *Il1r1* floxed alleles by Vav-iCre in both Vwf⁺ and Vwf⁻ HSC subsets (Supplementary Fig. 5c). Contrary to what we observed in germline *Il1r1*^−/−^ mice (Fig. 3e, f), *Il1r1*^FL/FL^ *Vav*-iCre^Tg/+^ mice showed equally efficient anti-GPIbα-induced cell cycle activation of Vwf⁺ HSC as in Wt mice (Fig. 3i). Together, these results demonstrate that direct IL-1R signaling through HSCs or other hematopoietic cells is not involved in the distinct cell cycle activation of HSCs in response to platelet depletion and rather implicate a role for non-hematopoietic IL-1 signaling in this process.

## Platelet depletion results in activation of IL-1 signaling in perivascular cells

Our studies in *Il1r1*^FL/FL^*Vav*-iCre^Tg/+^mice and RNA-sequencing analysis of BM niche cells suggested that IL-1 signaling in niche cells, rather than HSCs or other hematopoietic cells, might play a role in the feedback activation of HSCs after platelet depletion. Among niche cells, CBM-PV cells showed the highest transcriptional levels *of Il1r1* expression (Fig. 4a) and also showed distinct IL-1R protein expression, whereas all other stromal/endothelial cell populations were virtually negative for detectable IL-1R cell surface expression (Fig. 4b). In fact, almost all IL-1R expression in CBM could be assigned to Lepr⁺ PV cells (Fig. 4c), which expressed higher levels of critical HSC regulators such as *Cxcl12* and *Kitl*, in comparison with endosteal IL-1R^−/Lo^ BL-PV cells (Supplementary Fig. 3c). Importantly, the comparison of DE genes in CBM-PV cells from Wt and *Il1r1*^−/−^ mice in homeostasis and after GPIbα-

treatment revealed that the majority of DE genes identified in Wt mice following platelet depletion (Figs. 2f and 3b) were not differentially expressed in CBM-PV cells from platelet-depleted *Il1r1*^−/−^ mice (Fig. 4d), including the genes directly associated with the activation of the IL-1 signaling pathway (Fig. 4e). In addition to IL-1 signaling related genes, we found evidence for IL-1R-dependent DE of pathways implicated in extrinsic regulation of HSC proliferation, including the TGFβ signaling pathway (Figs. 2g and 4d, f, g)[34]. This included up-regulation of *Vasn* (Vasorin), a direct inhibitor of TGFβ signaling[35] previously implicated in HSC activation[36]. Other IL-1R-dependent DE genes included *Fst* (Follistatin), a direct inhibitor of BMP/Activin signaling[37] and previously shown to inhibit Mk differentiation from hematopoietic progenitor cells[38], and genes associated with cell adhesion and integrin binding/ regulation (Figs. 2f, g and 4d–g).

## IL-1 signaling in perivascular cells is critical for optimal activation of Vwf⁺ HSC

To more directly investigate the role of IL-1 signaling in CBM-PV cells in the activation of Vwf⁺ HSCs in response to thrombocytopenia, we induced platelet depletion in mice with conditional deletion of *Il1r1* specifically in Lepr⁺ PV cells (Supplementary Fig. 5d). *Il1r1*^FL/FL^*Lepr*-Cre^Tg/+^ mice showed a significant reduction in actively cycling (S-G2-M) Vwf⁺ HSCs, in comparison to anti-GPIbα-treated control mice (Fig. 5a). The reduction in cell cycle activation was comparable to the one observed in germ-line deleted *Il1r1*^−/−^ mice (Fig. 3e, f), supporting that Lepr⁺ CBM-PV cells are the main cells involved in IL-1R-dependent activation of Vwf⁺ HSCs in response to anti-GPIbα-induced thrombocytopenia. Corroborating these findings, administration of recombinant IL-1 to *Il1r1*^FL/FL^*Lepr*-Cre^Tg/+^ mice resulted in reduced recruitment of Vwf⁺ HSCs into active S-G2-M (Fig. 5b). Moreover, specific deletion of *Il1r1* in Lepr⁺ PV cells resulted in a small but significant delay in platelet recovery post platelet depletion (Fig. 5c).

To gain further insight into how Vwf⁺ HSCs are recruited into proliferation in response to anti-GPIbα-induced platelet depletion we performed RNA-sequencing of Vwf⁺ and Vwf⁻ HSCs. RNA-sequencing of Vwf⁺ HSCs identified 230 differentially expressed genes after platelet depletion, of which 170 were exclusively differentially expressed in Vwf⁺ and not Vwf⁻ HSCs, and 24 of these already distinguished Vwf⁺ and Vwf⁻ HSCs in homeostasis, including platelet/Mk lineage associated genes (Fig. 5d, e, Supplementary Fig. 5e, f and Supplementary Data 6, 7). Comparison of Vwf⁺ and Vwf⁻ HSCs from platelet-depleted mice revealed enrichment of cell cycle activation genes in Vwf⁺ HSCs, confirming at the molecular level the preferential activation of Vwf⁺ HSCs in response to platelet depletion (Supplementary Fig. 5g and Supplementary Data 8). Genes differentially expressed in Vwf⁺ HSCs after platelet activation and depletion are mainly associated with integrin signaling and cell adhesion, known to be regulated by TGFβ signaling[39], but also cell cycle, blood coagulation and response to stress/inflammation (Fig. 5d–f). While we cannot exclude the involvement of other signaling pathways in the regulation of these genes, in line with the down-regulation of *Fst* in CBM-PV cells (Fig. 4g), the candidate Activin/ BMP target genes *Runx3* and *Id1*[40] were respectively down- and up-regulated in Vwf⁺ but not Vwf⁻ HSCs post platelet depletion (Fig. 5g). Of note, the gene for the α1,6-fucosyltransferase (*Fut8*) which

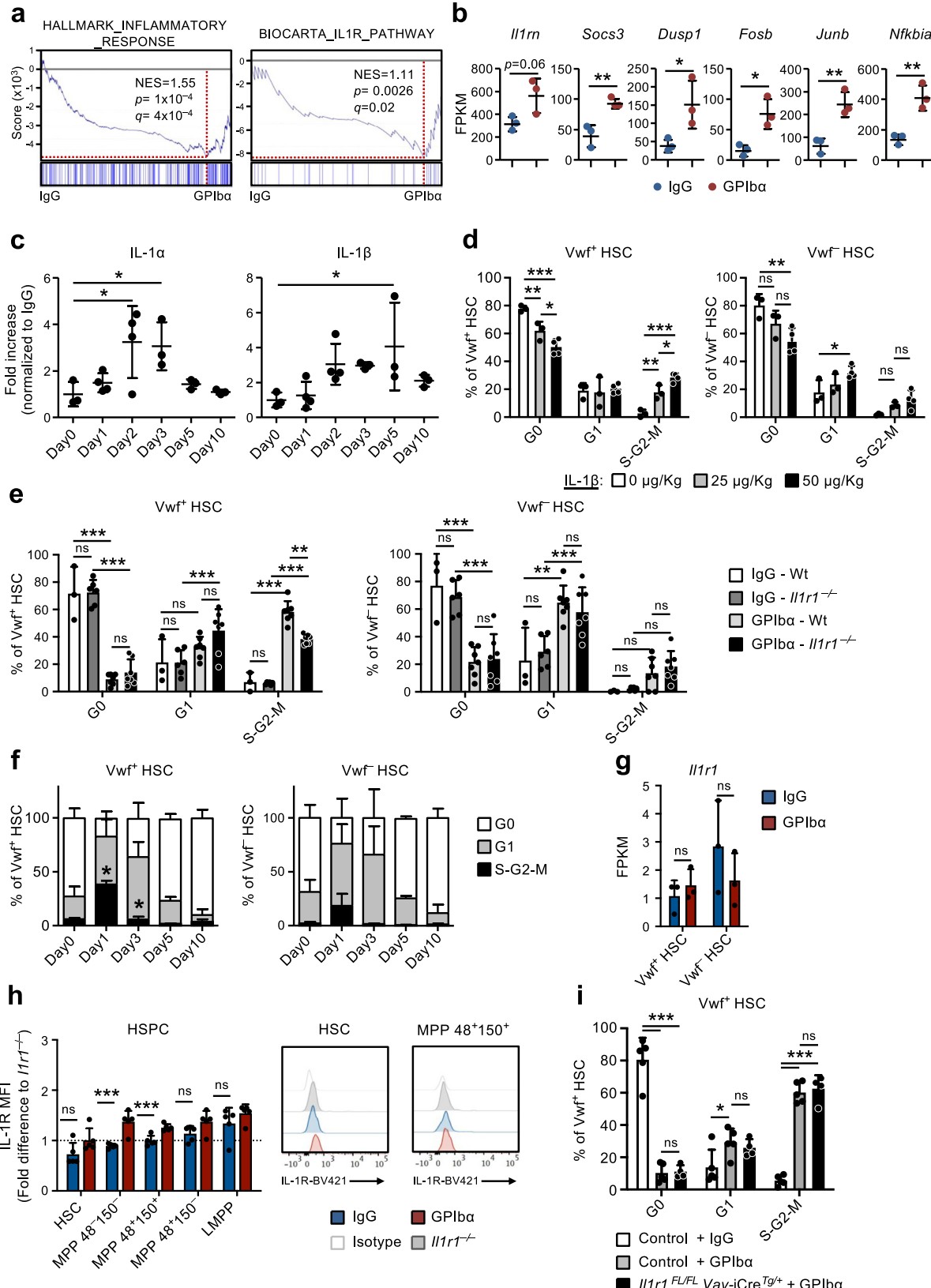

differentially regulates the activity of the TGFβ and Activin receptors[41] and is also a regulator of integrin signaling[42] was found up-regulated specifically in Vwf⁺ HSCs post platelet depletion. In addition, in response to anti-GPIbα-induced platelet depletion Vwf⁺ HSCs but not Vwf- HSCs up-regulated *Fkbp1a*, an intracellular regulator of Smad signaling that modulates the intensity and duration of the signals

downstream of the TGFβ, Activin and BMP receptors[43,44] (Fig. 5g). Together with the down-regulation of *Fst* in CBM-PV cells (Fig. 4g), these data suggest that a shift from quiescence-reinforcing TGFβ signaling to activating Activin/BMP signaling may lead to alterations in integrin activity, cell adhesion and cell cycle status of HSCs in response to acute platelet activation and depletion.

**Fig. 3 | Hematopoietic-extrinsic IL-1 signaling is critical for activation of platelet-biased HSCs in response to acute platelet depletion.** (Related to Supplementary Fig. 5). **a** Gene set enrichment analysis (GSEA) of global gene expression data from CBM-PV cells for the indicated gene sets. NES, normalized enrichment score (or scaled score). **b** Expression of IL-1 signaling pathway affiliated genes in CBM-PV cells 1 day post-platelet depletion. Data represent mean ± SD FPKM of 3 biological replicates from 2 independent experiments. **$p < 0.01$; *$p < 0.05$ (two-sided t-test). **c** Mean ± SD levels of IL-1α and IL-1β in bone marrow extracellular fluid isolated from mice at the indicated time points post platelet depletion (GPIbα antibody treatment). Control mice (Day 0) received isotype (IgG) control antibody. Data are from 3 (Day0), 4 (Day1), 4 (Day2), 3 (Day3), 3 (Day5) and 3 (Day10) mice from 4 independent experiments. *$p < 0.05$ (1-way ANOVA with Dunnett's multiple comparisons). **d** Cell cycle analysis of Vwf⁺ (left) and Vwf⁻ (right) HSCs from mice 1 day post intravenous administration of the indicated amounts of IL-1β. Data are mean ± SD of 3 mice receiving 0 or 25 µg/Kg, and 4 mice receiving 50 µg/Kg IL-1β, in 2 independent experiments. Cell cycle analysis of Vwf⁺ (left) and Vwf⁻ (right) HSCs from Wt and $Il1r1^{-/-}$ mice 1 day (**e**) or for HSCs from $Il1r1^{-/-}$ mice at the indicated time points (**f**) post platelet depletion. **e** Mean ± SD data from 3 (IgG-Wt), 6 (IgG-$Il1r1^{-/-}$) 7

(GPIbα -Wt) and 7 (GPIbα -$Il1r1^{-/-}$) mice from 5 independent experiments. **f** Mean ± SD frequencies from 6 (Day0), 7 (Day1), 3 (Day3), 3 (Day5) and 3 (Day10) mice in 4 independent experiments. *$p < 0.05$ (in comparison to same analysis of Wt Vwf⁺ and Vwf⁻ HSCs in Fig. 1d). **g, h** $Il1r1$/IL-1R expression analysis (**h**) at RNA level by RNA-sequencing (FPKM) in Vwf⁺ and Vwf⁻ HSCs and (**i**) at protein level by flow cytometry in HSPCs subsets, isolated from mice in homeostasis or 1 day post platelet depletion. **h** Mean ± SD FPKM data of 3 biological replicates per condition. **i** Mean ± SD data of Mean fluorescence intensity (MFI) normalized to the MFI of the equivalent cell population in $Il1r1^{-/-}$ mice analyzed within the same experiment. Data are from 5 mice per condition, in 2 independent experiments. **i** Cell cycle analysis of Vwf-tdTomato⁺ HSCs from mice with conditional deletion of $Il1r1$ in all hematopoietic cells ($Il1r1^{FL/FL}$ Vav-Cre$^{Tg/+}$) 1 day post platelet depletion. Controls include Vwf-tdTomato⁺ HSCs from $Il1r1^{FL/+}$ Vav-Cre$^{Tg/+}$, $Il1r1^{+/+}$ Vav-Cre$^{Tg/+}$ and Vav-Cre$^{+/+}$ mice (representing genotypes without an IL-1R loss of function). Data represent mean ± SD frequencies of 5 (control-IgG), 5 (control-GPIbα) and 4 ($Il1r1^{FL/FL}$ Vav-Cre$^{Tg/+}$-GPIbα) mice from 3 independent experiments. ***$p < 0.001$; **$p < 0.01$; *$p < 0.05$; ns non-significant ($p > 0.05$); using two-sided t-test (**b, g, h**) or 2-way ANOVA with Tukey's multiple comparisons (**d–f, i**). See also Supplementary Fig. 4.

## Platelet activation is essential for the HSC response to platelet depletion

The gene expression analysis of CBM-ECs revealed several genes up-regulated upon platelet depletion that are associated with platelet activation and coagulation (Fig. 2f, g). This is in line with the mechanism by which platelets are frequently consumed[45], here mimicked by GPIbα-mediated platelet depletion, which functions by inducing Fc-independent platelet activation, leading to platelet siali-dase neuraminidase-1 (NEU) translocation to the membrane, desialy-lation and subsequent clearance in the liver[20]. Moreover, platelets are recognized as important mediators of inflammation[46,47] and rapidly upregulate IL-1 protein expression upon activation (Fig. 6a), as previously shown by others[48–50]. This supports a role for platelet activation and subsequent consumption in mediating the herein observed IL-1-dependet activation of Vwf⁺ HSCs in response to thrombocytopenia. To further investigate this possibility we treated mice with NEU[51], which similarly to the GPIbα-treatment (Fig. 1a) leads to efficient platelet depletion (Fig. 6b) but bypasses platelet activation[20]. While GPIbα-treatment efficiently activated platelets in vitro, as measured by surface P-Selectin (CD62P) staining, in vitro NEU treatment resulted only in a very mild platelet activation, and only at high concentrations (Fig. 6c). In vitro NEU activity was however confirmed by *Ricinus communis* agglutinin I (RCA-1) labeling (Fig. 6d), which specifically binds to desialylated proteins[20]. Despite depleting platelets in vivo with the same efficiency, unlike anti-GPIbα treatment NEU treatment did not result in HSC cell cycle activation (Fig. 6e) nor did it sig-nificantly increase the numbers of Vwf⁺ or Vwf⁻ HSCs (Fig. 6f), LSK Flt3⁻CD48⁺CD150⁺ MPPs, pre-MegE and MkPs in BM (Supplementary Fig. 6a, b). Of note, in NEU treated mice the levels of IL-1α and IL-1β in BM remained largely unchanged (Fig. 6g). These results suggest that rather than the mere loss of platelets, anti-GPIbα activation-induced depletion of platelets is involved in the observed IL-1-dependent acti-vation of HSCs. Previous studies implicated a role for Mks and their secreted factors in the regulation of HSC quiescence/proliferation[31–33,52]. Since Mks also express GPIbα, to more specifically demonstrate a role of platelets in the observed HSC activation in response to anti-GPIbα antibody treatment, independently of Mks, we administered the anti-GPIbα antibody to mice in which platelets had been efficiently depleted with NEU-treatment (Fig. 6b), which does not alter the number of Mks[33]. This sequential NEU-GPIbα treatment resulted in reduced cell cycle activation of Vwf⁺ HSCs, when compared to GPIbα treatment alone (Fig. 6h), confirming the involvement of platelets in this process. However, the fact that some HSC activation was observed in mice with NEU-depleted platelets, when compared to control mice with normal platelet numbers (Figs. 1d and 3e), also

supports a role of Mks in the observed HSC activation in response to anti-GPIbα antibody treatment.

Platelets store multiple inflammatory modulators in platelet granules, which are released upon platelet activation[53,54]. To further investigate if the release of platelet granule contents is required for GPIbα-mediated HSC activation we induced platelet depletion (GPIbα treatment) in mice deficient for *Nbeal2* ($Nbeal2^{-/-}$), which lack platelet α-granules[55]. $Nbeal2^{-/-}$ mice have overall normal hematopoiesis despite a small decrease in platelet numbers and a slight increase in HSCs (Supplementary Fig. 6c–e). One day following anti-GPIbα-treatment $Nbeal2^{-/-}$ mice had a significantly reduced frequency of actively cycling HSCs when compared to Wt mice, despite of a higher frequency of HSCs being in cycle prior to treatment (Fig. 6i). Together, these find-ings suggested that IL-1 and other regulators secreted upon platelet activation may play a role in regulating the activation of HSCs in response to thrombocytopenia.

## Discussion

Here we explored and unraveled a niche-dependent feedback loop by which normally quiescent HSCs in the central BM can be rapidly acti-vated in response to acute peripheral thrombocytopenia induced by activation-mediated elimination of platelets (Fig. 6j). A systematic transcriptional profiling of different niche cells and HSCs in the BM of mice in homeostasis and after acute platelet depletion established that Lepr⁺ CBM-PV niche cells in the BM are critical cellular components of this feedback loop. Moreover, the lack of HSC activation when platelet depletion was induced by NEU, bypassing the activation-dependent step of normal platelet consumption, established a role for IL-1 pro-duced by platelets activated upon consumption, as a feedback med-iator from peripheral platelets to Lepr⁺ PV niche cells in the central BM. Similarly to NEU, other platelet antibodies frequently observed in ITP patients but that lead to platelet clearance by opsonization rather than activation (e.g GPIIb/IIIa and GPIbIX) are not expected to induce the feedback mechanism we described here. Of note, a recent study using a different method to deplete platelets did not report major changes in HSCs proliferation and downstream differentiation[56], probably reflecting that, as reported here, the activation of platelets while being consumed and not their mere absence, is critical for the HSC response to thrombocytopenia.

Biotin and H2B-mCherry labeling-dilution experiments provided definitive functional evidence that activation-induced platelet deple-tion results in recruitment of quiescent LT-HSCs into active prolifera-tion. Although several lines of data demonstrated that cell cycle activation was preferentially induced within the Vwf⁺ HSC compart-ment, previously shown to contain platelet-biased HSCs[9,18], Vwf⁻ HSCs

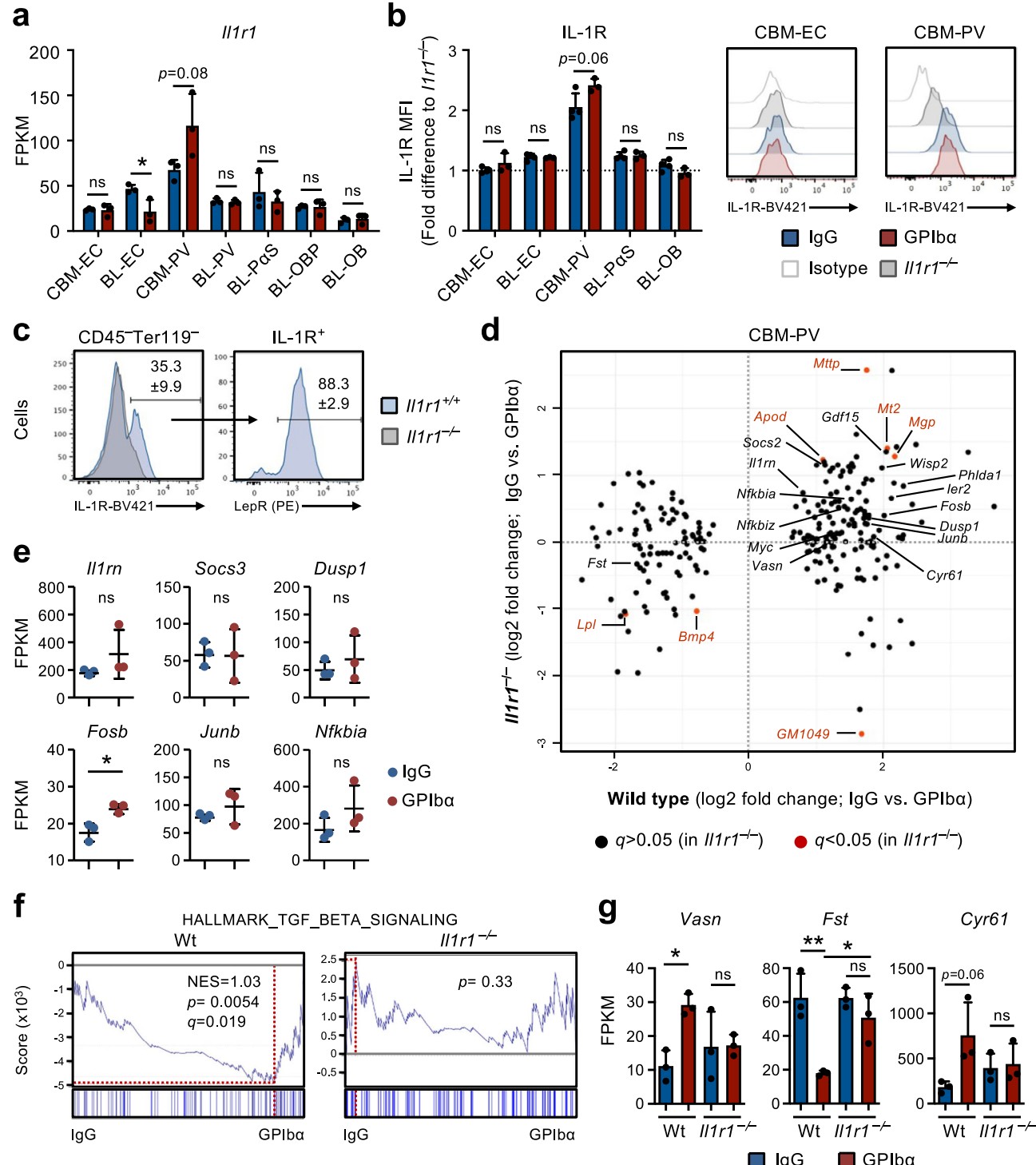

**Fig. 4 | IL1R expression defines a population of perivascular bone marrow stromal cells implicated in the HSC response to platelet depletion. a** RNA-sequencing analysis of *Il1r1* gene expression (FPKM) in different niche cells isolated from mice in homeostasis (IgG treated) or 1 day post platelet depletion (GPIbα treated). Mean ± SD FPKM data of 3 biological replicates from 2 independent experiments. **b**, **c** Flow cytometric analysis of IL-1R expression in different endo-thelial/stromal cell populations isolated from mice in homeostasis or 1 day post platelet depletion. Mean ± SD data of Mean fluorescence intensity (MFI) normalized to the MFI of the equivalent cell population in *Il1r1⁻/⁻* mice analyzed within the same experiment (**b**). **c** Frequency of Lepr⁺ PV cells in total IL-1R⁺ CBM non-hematopoietic cells isolated from mice in homeostasis. Data from 4 (IgG) and 3 (GPIbα) mice in 2 independent experiments. **d** RNA-sequencing analysis of CBM-PV cells isolated

from *Il1r1⁺/⁺* and *Il1r1⁻/⁻* mice in homeostasis and after platelet depletion, for the expression of CBM-PV-GPIbα treatment responsive genes. Data from 3 biological replicates per condition. **e** Expression of IL-1 signaling pathway affiliated genes in CBM-PV cells isolated from *Il1r1⁻/⁻* mice 1 day post platelet depletion. Mean ± SD FPKM data of 3 biological replicates per condition. GSEA of global gene expression data for the indicated gene set (**f**) and expression (FPKM; Mean ± SD) of the indi-cated genes (**g**), in CBM-PV cells from wild type and *Il1r1⁻/⁻* mice in homeostasis and after platelet depletion. Data from 3 mice per condition. NES, normalized enrich-ment score (or scaled score). *$p < 0.05$; **$p < 0.01$; ns, non-significant ($p > 0.05$); using two-sided t-test (**a**, **b**, **e**) and 2-way ANOVA with Tukey'a multiple compar-isons (**g**).

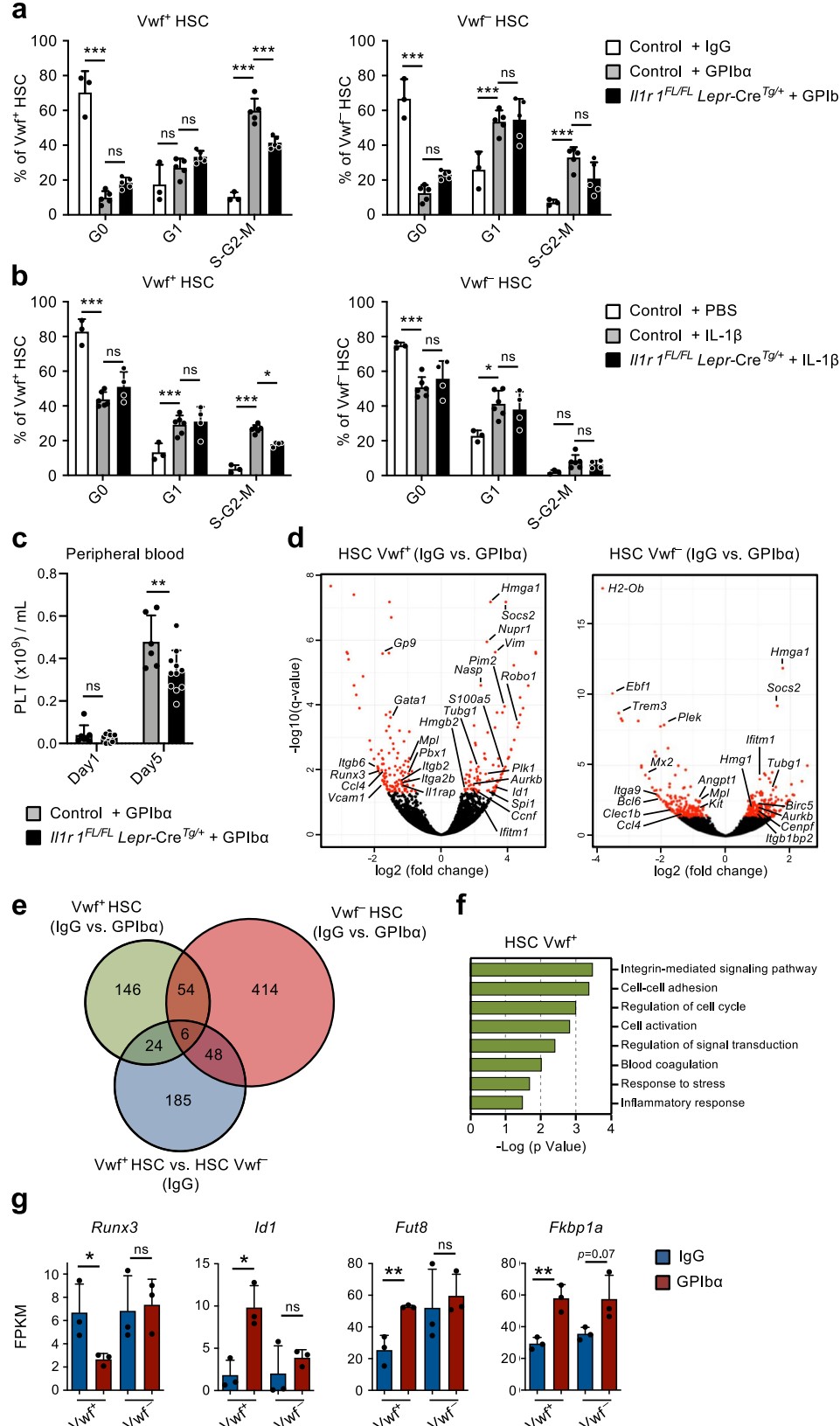

were also activated to some extent. While the specific cell cycle analysis showed preferential activation of the Vwf⁺ HSC compartment, Biotin labeling-dilution was observed in both Vwf⁺ and Vwf⁻ compartments. This difference could reflect the hierarchical relationship previously established to exist between Vwf⁺ and Vwf⁻ HSCs, with Vwf⁺ HSCs giving rise to Vwf⁻ HSCs and not vice versa[9]. In addition, while

Vwf⁺ HSCs numbers increase following platelet depletion, an initial reduction in Vwf⁻ HSC numbers is observed, which might be explained by downstream differentiation of Vwf⁻ HSCs. Regardless, the consequence of the platelet depletion and activation is a broad activation of HSCs. In line with this, while the activated HSCs effectively replenished platelets upon transplantation they were equally effective at

**Fig. 5 | IL-1-dependent activation of perivascular bone marrow stromal cells enhance Vwf + HSC activity in response to platelet depletion.** (Related to Supplementary Fig. 6). **a–c** Analysis of mice with conditional deletion of *Il1r1* in Lepr[+] perivascular cells (*Il1r1*[FL/FL] *Lepr*-Cre[Tg/+]) mice after platelet depletion. Controls include *Il1r1*[FL/+] *Lepr*-Cre[Tg/+], *Il1r1*[+/+] *Lepr*-Cre[Tg/+], *Il1r1*[FL/FL] *Lepr*-Cre[Tg/+] (IgG only) and *Lepr*-Cre[+/+] mice. **a** Cell cycle analysis of Vwf[+] and VWF[−] HSCs 1 day post platelet depletion. Data represent mean ± SD frequencies of 3 (control-IgG), 5 (control-GPIbα) and 5 (*Il1r1*[FL/FL] *Lepr*-Cre[Tg/+]-GPIbα) mice from 3 independent experiments. **b** Cell cycle analysis of Vwf[+] and VWF[−] HSCs from mice 1 day post intravenous administration of 50ug/Kg of IL-1β. Data are mean ± SD of 3 (control-IgG), 6 (control-GPIbα) and 4 (*Il1r1*[FL/FL] *Lepr*-Cre[Tg/+]-GPIbα) mice from 3 independent experiments. **c** Peripheral blood analysis of platelet recovery at the indicated time points post platelet depletion. Mean ± SD platelet numbers from 6 (control) and 11 (*Il1r1*[FL/FL] *Lepr*-Cre[Tg/+]) mice in 3 independent experiments. **d–g** Differential gene expression analysis in Vwf[+] and Vwf[−] HSCs isolated from wild type mice 1 day post platelet depletion (GPIbα treatment). **d** Volcano plots of genes differentially expressed in Vwf[+] and Vwf[−] HSCs. Red dots indicate genes with significant expression differences (adjusted *p* value (*q*)<0.05). **e** Venn diagram showing number of differentially expressed genes in Vwf[+] and Vwf[−] HSCs post platelet depletion and between these HSC subsets in homeostasis (IgG). **f** GO terms analysis of biological processes/pathways up-regulated in Vwf[+] HSCs after platelet depletion. **g** Expression (FPKM) of genes associated with Activin/BMP signaling in Vwf[+] and Vwf[−] HSCs from Wt mice in homeostasis and 1 day post platelet depletion. All data (**d–g**) represent mean ± SD FPKM data of 3 biological replicates per genotype and condition. *$p < 0.05$; **$p < 0.01$; ***$p < 0.001$; ns non-significant ($p > 0.05$); using 2-way ANOVA with Tukey's multiple comparisons (**a, b**), 2-way ANOVA with Sidak's multiple comparisons (**c**) and two-sided t-test (**g**). See also Supplementary Fig. 5.

replenishing other myeloid cells. This is in agreement with most HSCs having the capacity to potently replenish all blood cell lineages[18]. Collectively, this suggests that activation-induced platelet depletion results in broad activation of quiescent HSCs, all of which possess extensive potential for platelet replenishment. In fact, whereas subsets of HSCs have been shown to not replenish all blood cell lineages, platelets is the only blood cell lineage which all (Vwf[+] and Vwf[−]) LT-HSCs contribute actively to upon transplantation[18]. Our findings establish that the activation of platelets is critical for the optimal feedback activation of quiescent HSCs in response to thrombocytopenia, uncovering a new level of regulation by which the activation of HSCs is triggered by the specific consumption of a terminally differentiated blood cell lineage.

Whereas IL-1 has previously been implicated to directly promote myeloid lineage output of HSCs[30,57], the IL-1-dependent activation of HSCs following platelet depletion was not mediated through a direct effect of IL-1 on HSCs or other hematopoietic cells, since the HSC activation was not affected by a pan-hematopoietic deletion of IL-1R expression, including complete deletion in HSCs. Rather we demonstrate that IL-1-dependent activation of HSCs is mediated through IL-1R signaling in Lepr[+] CBM-PV niche cells, as demonstrated through specific IL-1R deletion in Lepr[+] cells. The fact that deletion of IL-1R expression specifically in Lepr[+] PV cells leads to comparable impairment in HSC activation post platelet depletion as a germ-line deletion of *Il1r1*, suggests that Lepr[+] PV cells are the critical BM target cells for the IL-1R dependent signaling in the observed feedback loop. This was further supported by IL-1R expression in the BM niche being largely restricted to Lepr[+] CBM-PV cells, while Lepr[+] BL-PV cells expressed much lower levels of IL-1R. Moreover, Lepr[+] CBM-PV cells specifically demonstrated up-regulation of components of the IL-1 signaling pathway, in an IL-1R-dependent manner, in response to acute activation-dependent platelet depletion. Our findings are in agreement with previous studies implicating a critical role of Lepr[+] CBM-PV cells as HSC niche cells[58,59]. We also confirmed that Lepr[+] CBM-PV cells express higher levels of known HSC regulators such as *Cxcl12* or *KitL*, than the other candidate and IL-1R negative BM niche cell populations, including the Lepr[+] endosteal BL-PV cells. Upon platelet depletion IL-1R signaling in Lepr[+] CBM-PV cells results in the up-regulation of direct inhibitors of TGFβ signaling (*Vasn* and *Wisp2*)[35,60] and down-regulation of a direct inhibitor of BMP/Activin signaling (*Fst*)[37]. This is compatible with the observed cell cycle activation of HSCs resulting from a shift in Smad signaling in these cells, leading to the inhibition of quiescence-reinforcing TGFβ signals[32,61–63] and the activation of BMP/Activin signaling[34], which was previously implicated in promoting Mk-lineage differentiation[38].

*Il1r1* deficiency did not lead to a complete abrogation of the observed HSC cell cycle activation post platelet depletion and activation, implicating additional factors in this process. These may include TNFα, which was increased in the BM extracellular fluid in a similar manner as IL-1, and THPO, which we have previously shown to be increased in the serum of mice after platelet depletion[9]. Moreover, we observed increased levels of PF4 in the BM extracellular fluid of mice after platelet depletion. However, PF4 has been previously implicated as an HSC quiescence rather than activation inducing factor[31] so it is unlikely to be involved in the activation of HSCs post platelet depletion. Instead, the increased levels of PF4 may participate in a negative feedback mechanism to rapidly bring HSCs back to quiescence, as observed in our studies. While the increased levels of PF4 may derive from BM Mks[31], we also observed increased *Pf4* expression in CBM-ECs. Despite the lower levels of *Pf4* expression in CBM-ECs when compared to Mks, ECs are considerably more abundant in BM where they are incorporated into HSC niches and therefore may represent a relevant source of PF4 for HSCs in situations of perturbed platelet homeostasis. We did not observe changes in BM extracellular fluid levels of TGFβ1 and FGF-1, which have been implicated in the regulation of HSC quiescence/proliferation[32], failing to support that they are involved in the activation of HSC proliferation post platelet depletion and activation. However, we cannot exclude that the levels of these and other factors may only change locally in the HSC niche, which might not be detectable when analyzing total BM extracellular fluid. Regardless, further genetic studies will be required to establish the role of additional regulators in the IL-1-independent HSC proliferation following platelet activation and depletion, and whether they act through cells in HSC niches.

The effect of in vivo IL-1 administration on HSC proliferation was only partially abrogated by conditional deletion of *Il1r1* in CBM-PV cells. This might in part reflect that we did not observe complete recombination and loss of IL-1 receptor expression in Lepr[+] PV cells in *Il1r1*[FL/FL] *Lepr*-Cre[Tg/+] mice, in line with previous reportes of inefficient *Lepr*-Cre mediated recombination of other floxed alleles[58], but could also reflect that systemic administration of IL-1 might in part activate HSCs through other IL-1R expressing cells.

We found very low or undetectable levels of IL-1α and IL-1β expression in steady-state BM endothelial and mesenchymal cell populations, as previously reported[30], and none of the niche cell populations up-regulated either IL-1 isoforms after platelet depletion. In contrast, despite being enucleated, platelets contain unprocessed RNAs such as *Il1b* and a functional, activation-dependent spliceosome and translation machinery[49,50], facilitating production of de novo IL-1 upon platelet activation[48]. Previous studies showed increased levels of IL-1 in serum post GPIbα-mediated platelet depletion[48,64] and we now show that this increase extends to the serum specifically harvested from the BM. We also found expression of both IL-1 isoforms in BM Mks.

A number of previous studies have implicated the role of platelets[9,16,17] as well as of Mks[31–33,52] in regulation of HSCs. However, the genetic approaches used to implicate the involvement of platelets or Mks have most likely affected both cell types, by using targets shared by Mks and platelets. NEU efficiently and in an activation-independent manner depletes platelets but not Mks[31]. By sequentially administrating NEU prior to the GPIbα antibody, we were able to more specifically

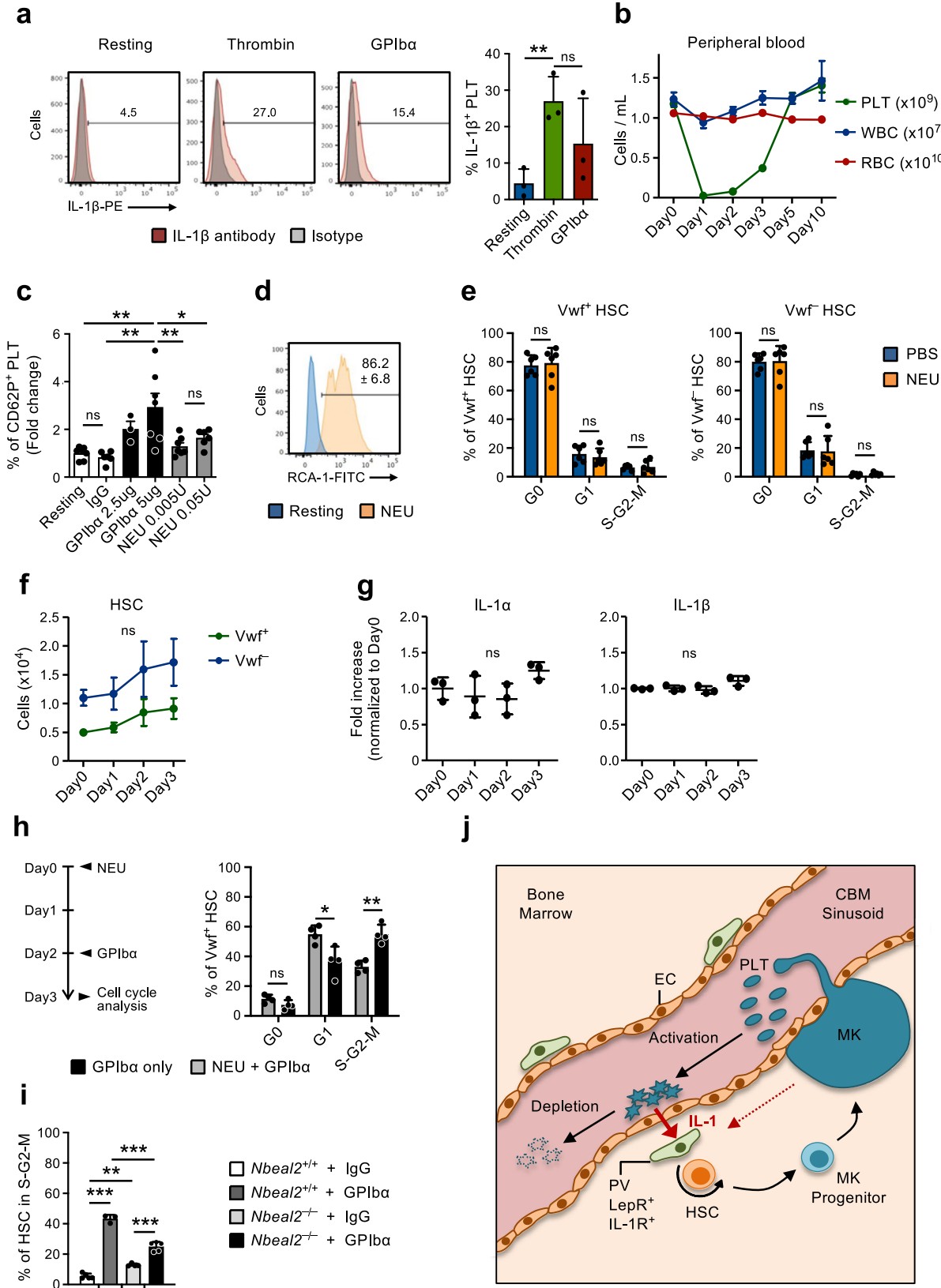

investigate the effect of GPIbα antibody treatment in the absence of platelets but with normal Mk numbers, and therefore, determine the specific contribution of platelets, independently of Mks. The reduced recruitment of Vwf⁺ HSCs into proliferation in this setting supports a specific role for platelets in this process. However, the fact that the absence of platelets did not fully abrogate the observed Vwf⁺ HSC

activation also supports a role of Mks in the observed HSC activation in response to the GPIbα antibody treatment, and as an additional relevant source of IL-1 in the observed feedback loop, in line with previous studies[64,65] (Fig. 6j).

Platelets also store several inflammatory modulators in cytoplasmatic granules and vesicles, which upon activation are either released

**Fig. 6 | Neuraminidase-mediated platelet depletion does not activate HSCs.** (Related to Supplementary Fig. 4). **a** Expression of IL-1β in platelets resting or after in vitro activation (3hrs) with thrombin or GPIbα antibody. (Left), representative FACS profiles. Numbers in plots are average frequencies from 3 independent experiments. (Right) Frequency of IL-1β+ platelets; mean ± SD of 3 biological replicates per condition in 3 independent experiments. Each biological replicate consists of platelets pooled from 2-3 mice. **b** Kinetics analysis of peripheral blood cell lineages following in vivo Neuraminidase (NEU) administration. Data represent mean ± SEM of 10 (Day0), 7 (Day1), 8 (Day2), 9 (Day3), 6 (Day5) and 3 (Day10) mice from 6 independent experiments. PLT platelets, WBC white blood cells, RBC red blood cells. **c** Expression of surface P-selectin (CD62P) on platelets measured by flow cytometry after in vitro incubation with GPIbα antibody or NEU, at the indicated concentrations. Data represent mean ± SD fold changes of % CD62P+ cells in each condition in relation to untreated (resting) platelets, of 7 (Resting), 6 (IgG), 3 (GPIbα-2,5ug), 7 (GPIbα-5ug), 7 (NEU 0.005U) and 6 (NEU 0.05U) mice in 3 independent experiments. **d** In vitro neuraminidase (NEU) activity in resting platelets or after 30 min treatment with NEU, analyzed by RCA-1 binding. Representative profile from 1 out of 3 biological replicates. Numbers indicate mean ± SD % RCA-1+ platelets. **e** Mean ± SD cell cycle phase distribution of Vwf+ (left) and Vwf- (right) HSCs 1 day post platelet depletion with NEU. Data from 6 mice per condition, in 3 independent experiments. **f** FACS-based assessment of the HSC compartment in bone marrow of mice at the indicated time points after platelet depletion with NEU. Data represent absolute numbers of *Vwf*-GFP+ (Vwf+) or *Vwf*-GFP- (Vwf-) HSCs

(average ±SEM) at the indicated time points after platelet depletion. Data are from 5 (Day0), 4 (Day1), 5 (Day2) and 6 (Day3) mice in 4 independent experiments. No significant changes were observed in numbers of Vwf+ or Vwf- HSCs at any time point. **g** Mean ± SD levels (fold-increase relative to Day0) of the indicated cytokines in bone marrow extracellular fluid isolated from mice at the indicated time points post platelet depletion with Neuraminidase. Data from 3 mice per time point in 2 independent experiments. **h** Mice were treated with NEU at day 0, followed by GPIbα antibody administration at day 2 and analyzed at day 3 (left) for cell cycle phase distribution in Vwf+ HSCs (right). Control mice were treated only with GPIbα antibody and analyzed 1 day later. Data represent mean ± SD frequencies of 4 mice per group in 2 independent experiments. **i** HSCs in S-G2-M in *Nbeal2*−/− mice 1 day post platelet depletion (GPIbα). Data represent mean ± SD cell frequencies of 5 (Wt-IgG), 3 (Wt-GPIbα), 4 (*Nbeal2*−/−-IgG) and 5 (*Nbeal2*−/−-GPIbα) mice per condition from 3 independent experiments. **j** Scheme depicting the feedback mechanism proposed. While being consumed activated platelets secrete IL-1, which activates IL-1R expressing PV cells to induce HSC proliferation and differentiation toward the platelet lineage. As indicated, Mks may also contribute to the described recruitment of HSCs into proliferation in response to treatment with the anti-GPIbα antibody resulting in activation-dependent platelet depletion. For all data ***p < 0.001; **p < 0.01; *p < 0.05 (Only indicated for significant differences) using 1-way ANOVA with Tukey's multiple comparisons (**a**, **c**, **f**, **i**), 2-way ANOVA with Sidak's multiple comparisons (**e**, **h**) or Dunnett's multiple comparisons (**g**); ns non-significant. See also Supplementary Fig. 6.

or exposed at the cytoplasmatic membrane[53,54]. Platelet depletion in *Nbeal2*−/− mice, which lack α-granules in platelets[55], resulted in reduced HSC cell cycle activation, further establishing the importance of platelet-derived factors in HSC activation upon acute platelet depletion. An in-depth analysis of the platelet proteome should facilitate a more complete understanding of the role of release of different platelet-derived factors in the activation of HSCs.

While we did not specifically investigate this, in addition to HSCs, our findings are also compatible with the involvement of committed Mk progenitors such as stem-like Mk-committed progenitors[24], or Mk-repopulating progenitors (MkRPs) and Mk-erythrocyte repopulating progenitors (MERPs)[66], in the observed rapid response to platelet depletion and activation. That otherwise quiescent phenotypically and functionally defined HSCs are rapidly activated in a niche-dependent manner in response to platelet activation and depletion uncovers a novel mechanism by which the activation-induced depletion of a mature blood cell lineage in the periphery is accompanied by the production of regulators which feedback to a distinct stromal-niche cell in the central BM to promote the proliferation of quiescent HSCs with extensive potential for replenishing the entire lineage commitment pathway required for platelet replenishment (Fig. 6j).

## Methods

### Mice
All mice were bred and maintained in accordance with UK Home Office regulations. All procedures were performed under project licenses 30/3103 and P2FF90EE8 approved by the Oxford University Clinical Medicine Ethical Review Committee. *Vwf*-GFP[9], *Vwf*-TdTomato[18], Osx-GFP::Cre[67], *Il1r1*−/−[68], *Il1r1*FL/FL mice[69], *Vav*-Cre[70], *Lepr*-Cre[71] and *Nbeal2*−/−[55] mice have been previously reported. All mouse lines were back-crossed for at least 6 generations onto a C57BL/6 genetic background and littermate controls were used in all experiments. Mice of both sexes were used in all experiments.

### Analysis of blood cell parameters
Mouse blood parameters were analyzed using ABX PENTRA 60C+ (Horiba) or XP-300 (Sysmex) automated blood cell analyzers.

### Platelet depletion and in vivo treatments
Platelet depletion was induced by one intra-venous (IV) administration of the anti-GPIBα antibody (R300; Emfret Analytics) at 2ug/g body

weight at the indicated time points before analysis. Control mice received the IgG isotype control antibody C301 (Emfret Analytics). Alternative methods for platelet depletion included IV administration of the anti-GPIBα antibody NIT E[20] at the indicated dosages, or Neuraminidase from *C.perfringens* (Roche) at 0.1U/mouse. Efficient platelet depletion was confirmed for all mice used. IL-1β (PeproTech) was administered IV at 0.025 and 0.05ug/g body weight as indicated. For Biotin proliferation assays EZ-Link™ Sulfo-NHS-LC-LC-Biotin (ThermoFisher) was administered IV at 170ug/g body weight in combination with the anti-GPIBα antibody R300 for platelet depletion or IgG isotype control.

### Competitive bone marrow transplantation
For the transplantation experiments using the H2B-mCherry dilution model, 50 H2B-mCherry^High or H2B-mCherry^Low LSKFlt3−CD48−CD150+ bone marrow cells were purified by FACS from H2B-mCherry^Tg/Tg*Gata1*-GFP^Tg/+ (CD45.2) 3 days post platelet depletion and intravenously transplanted together with 2.5 ×10^5 unfractionated support/competitor CD45.1 bone marrow cells into CD45.1 (C57BL/6) recipient mice. For the transplantation experiments using the Biotin labeling-dilution model, 50 Vwf+Biotin^High or Vwf+Biotin^Low LSKFlt3−CD48−CD150+ bone marrow cells were purified by FACS from *Vwf*-GFP^Tg/+ (CD45.1) 2 days post platelet depletion and intravenously transplanted together with 2.5 ×10^5 unfractionated support/competitor CD45.2 bone marrow cells into CD45.2 (C57BL/6) recipient mice. In all experiments, 2 lethally irradiated (10.5 Gy, split dosage of 525 cGy each) recipient mice of more than 8 weeks of age were transplanted per donor and per population investigated. Flow cytometric analysis of CD45.1 and CD45.2 contribution to mature peripheral blood lineages was performed 16 weeks after transplantation. Donor contribution for the platelet/megakaryocyte lineage in peripheral blood was performed based on expression of the *Vwf*-GFP transgene[9,18].

### Bone marrow niche cells isolation
Long bones (femur, tibia and hip) were isolated and crushed in order to separate the bone/endosteal lining (BL) and central bone marrow (CBM) fractions. The bone fraction was then treated with Collagenase 1 (Worthington; 3 mg/mL, 2 × 45 min at 37 °C) and the CMB fraction with Collagenase 4 (Worthington; 2 mg/mL, 20 min at 37 °C). CBM fraction was then subjected to erythrocyte lysis with Ammonium chloride solution (Stem cell technologies) and CD45+ cells depletion using

mouse anti-CD45 beads and magnetic cell separation (MACS, Milteny Biotech). Single cell suspensions from both fractions were then stained with antibodies for further FACS analysis.

## Flow cytometry

Mouse BM single cell suspensions and peripheral blood cells were Fc-blocked and stained anti-mouse antibodies (described in Supplementary Table 1 and in the Reporting summary). Fluorescence-minus-one (FMO) controls, isotype control antibodies and negative populations were used as gate-setting controls. 7-Amino-Actinomycin D (7-AAD; Sigma) or 4′,6-diamidino-2-phenylindole (DAPI; Invitrogen) were used for dead cells exclusion. For HSC cell cycle analysis c-Kit+ cells were isolated from whole BM by magnetic separation (MACS; Milteny Biotech) and stained for cell surface markers prior to fixation and permeabilization with the BD Cytofix/Cytoperm Kit (BD Biosciences) for 30 min at 4 °C. Cells were then stained with Human anti-Ki67 or IgG antibody (BD Biosciences) overnight at 4ºC followed by DAPI staining for 1 h at 4 °C. FACS analyses were performed on BD LSRII or BD Fortessa X20 (BD Biosciences) and subsequent data analyses were performed with the FlowJo analysis software (TreeStar Inc). Absolute cell numbers were defined as cells per 2 legs (each including femur, tibia and hip bones). Cell sorting experiments were performed on BD FACS AriaIIu, AriaIII and Fusion cell sorters (BD Biosciences), with a mean cell sorting purity of $97.4 \pm 0.45$ % (mean ± s.e.m.). Single cell sorts were performed using the automatic cell deposition unit (ACDU). Single cell deposition efficiency was confirmed with fluorescent beads in all experiments (>99% of wells with 1 cell and no wells with more than 1 cell). FACS sorting of megakaryocytes was performed with a 2.0 scatter neutral density filter to decrease FSC signal intensity and allow visualization and gating of large cells, and with a 100 µm nozzle to increase cell viability.

## RNA-sequencing

Samples for RNA-sequencing (100 cells) were prepared using the SMARTer Ultra Low RNA kit for Illumina Sequencing (Clonetech), Nextera XT DNA Library Preparation Kit (Illumina) and Nextera XT Index Kit (24 Indexes, 96 Samples) (Illumina), and sequenced in a HiSeq2500 or HiSeq4000[72]. For data analysis, adapter removal was performed with Trimgalore (v.1.2.1, https://github.com/FelixKrueger/TrimGalore), alignment was performed to the mm10 mouse built with refSeq transcriptome annotation using tophat (version 2.0.10)[73]. Reads were sorted using samtools (version 0.1.19)[74,75], only primary alignments were used for subsequent steps. Read counting to trasncripts was performed using the packages Rsamtools (version 1.18.2) (http://bioconductor.org/packages/release/bioc/html/Rsamtools.html) and GenomicAlignments (version 1.2.1)[76] in R version (3.1.1). FPKM was calculated using the *DESeq2 fpkm* function. Differential expression analysis was performed using DESeq2 (version 1.18.1)[77] independently for each comparison. Principal component analysis was performed using the *prcomp* R function. GSEA analysis was performed against datasets from msigdb converted to mouse identifiers available from (http://bioinf.wehi.edu.au/software/MSigDB/) using liger (https://github.com/JEFworks/liger). GO Analysis was performed using the GOStats R package (version 2.44.0)[78]. Hierarchical clustering was generated based on Pearson correlation distance using the R function hclust and the 'ward.D2' method on the rlog transformed count values (calculated with DESeq2). Heatmaps were generated with Morpheus (Broad Institute). Venn diagrams were performed with BioVenn software[79]. Figure preparation was performed with the dendextend R package (version 1.8.0).

## Isolation of bone marrow extracellular fluid and cytokine analysis

Bone epiphyses were removed from the femurs and marrow was flushed out of the bone in 100ul PBS by centrifugation (300 g, 1 min).

Marrow from 2 femurs per mouse was resuspended, centrifuged (300 g, 3 min) and the supernatant frozen at −80 °C for posterior cytokine analysis. IL-1α, IL-1β, IL-6, IL-12p70, TNF and IFNγ Cytokine levels were investigated by flow cytometry using the cytokine beads array (BD Biosciences) according to manufacturer instructions.

The analysis of TGFβ1, PF4 and FGF1 levels in bone marrow extracellular fluid was performed by ELISA using the Human/Mouse/Rat/Porcine/Canine TGFβ1 Quantikine ELISA Kit, the Mouse PF4/CXCL4 Quantikine ELISA Kit and the Mouse FGF acidic/FGF1 DuoSet ELISA Kit according to manufacturer instructions (Bio-techne). In these experiments, bone marrow extracellular fluid was flushed out from 1 femur by centrifugation (3000 g, 1 min) in 50uL PBS.

## In vitro single cell Mk/GM differentiation assay

For the evaluation of lineage potential and timing of Mk cell emergence single Vwf+BiotinHi, Vwf+BiotinLo, Vwf−BiotinHi or Vwf−BiotinLo HSCs were sorted directly into 20uL X-Vivo15 medium (Lonza) supplemented with 10% fetal bovine serum (Sigma); 1% Pen/Strep (Gibco); $10^{-4}$M β-Mercaptoethanol (Sigma); mSCF (10 ug/mL; R&D Systems); hFlt3L (10 ug/mL; Immunex); hTHPO (10 ug/mL; PeproTech); mIL-3 (5 ug/mL; R&D Systems). Cultures were evaluated every day between days 2 and 8 for morphology and GFP expression using an inverted fluorescence microscope (Olympus). Fluorescent images were processed with Fiji software (ImageJ). Presence of Mk and myeloid (GM) cells was confirmed by May-Grunwald Giemsa (MGG)-stained cytospin preparations from selected wells (after 8 days of culture).

## Immunofluorescence

For whole-mount imaging of the sternum BM[80] vasculature was stained in vivo by IV administration of AF647-conjugated antibodies against CD31 and CD144 15 min before sternum isolation. The sternum central bone was longitudinally sectioned to expose the BM. Tissues were then fixed with paraformaldehyde (4%, 30 min, RT), permeabilized with TritonX-100 (0.5%, 1 h, RT), blocked with normal goat serum and stained with CD41-Bio and CD150-PE antibodies for 3 days, followed by 3 h staining with streptavidin-eF450 (See key resources table for antibodies references). Imaging was performed on a Zeiss 780 LSM multiphoton/confocal upright microscope. A Multi-photon MaiTai laser was used for the detection of collagen second harmonic signal (to identify bone). Image analysis was performed with Imaris 7.5.0 software (Bitplane). Mks were defined as cells >20 um, co-expressing *Vwf*-GFP, CD41 and CD150.

## In vitro platelet activation assays

Blood was withdrawn by heart puncture into anticoagulant citrate-dextrose solution (Sigma). Whole citrated blood was centrifuged (100 g, 10 min) in Tyrode's buffer (NaCl 134 mM, KCl 2.9 mM, $Na_2HPO_4.2H_2O$ 0.34 mM, $NaHCO_3$ 12 mM, HEPES 20 mM, $MgCl_2.6H_2O$ 1 mM; all from Sigma) pH6.5, to obtain platelet-rich plasma. Platelets were then washed twice (1400 g, 10 min) in Tyrode's buffer pH6.5, supplemented with the platelet activation inhibitors Apyrase (0.02 U/mL; Sigma) and Prostaglandin $E_1$ (0.05 µg/mL; Sigma). Washed platelets were resuspended in Tyrode's buffer pH7.35 supplemented with Glucose (1 mg/mL; Sigma) and $CaCl_2$ (1 mM; Fluka) at a concentration of $1-2 \times 10^9$ platelets/mL and left to recover (30 min, RT) before in vitro activation assay. For the in vitro activation assay $30-50 \times 10^6$ platelets per assay were incubated with anti-GPIbα antibody (R300, Emfret), Thrombin (Sigma) or Neuraminidase (Roche) at the indicated concentrations and times. Activation assay was stopped by fixation in paraformaldehyde (1%, 10 min, RT). Platelet activation was analyzed by staining with CD62P-PE antibody. In vitro Neuraminidase activity was confirmed by staining with fluorescein-labeled RCA-1 (Vector Labs). For IL-1β detection permeabilized platelets (0.1% Saponin, 10 min, RT; Sigma) were stained with anti-IL-1β antibody. See Supplementary Table 1 and Reporting Summary for antibody references.

### Analysis of *Vav*-Cre mediated deletion of the *Il1r1*^FL allele

Analysis of *Vav*-Cre mediated deletion efficiency of the *Il1r1*^FL allele was performed by Droplet digital PCR following whole genome amplification of DNA from 50 Vwf^+ or Vwf^- HSCs isolated from *Il1r1*^FL/FL *Vav*-Cre^Tg/+ mice. Whole genome amplification was performed with the REPLI-g single cell kit (Qiagen). In summary, 50 cells were directly FACS sorted into 0.2 ml PCR tubes (Thermo Scientific) containing 3 µl of Buffer D2, were according to manufacturer's instructions cells lysed with buffer D2 for 10 min at 65 °C, before adding stop solution. Lysed cells were incubated with reaction buffer and DNA polymerase (REPLI-g single cell Kit (Qiagen; provided with the kit) for 8 h at 30 °C, followed by 3 min incubation at 65°C to heat-inactive the DNA polymerase. Amplified DNA was diluted 1 to 33 in TE buffer (Invitrogen) before analysis. For Droplet Digital PCR analysis, a 20 µL PCR reaction mixture containing 1x ddPCR supermix for probes (no dUTP) (Bio-Rad), 1x primer-probe assay and 2.5 µl of the whole genome amplified and diluted DNA was partitioned into ~14,000–20,000 droplets per sample with Droplet Generation Oil for Probes (Bio-Rad). Droplets were prepared according to manufacturer's instructions on a QX200 droplet generator (Bio-Rad). Emulsified PCR reactions were run on a thermal cycler (Bio-Rad) incubating the plates at 95 °C for 10 min followed by 40 cycles at 94 °C for 30 s and 55 °C for 60 s, followed by 10 min incubation at 98 °C. The temperature ramp increment was 2.5 °C/s for all steps. Deleted alleles were detected using a FAM probe (Assay ID dMmuCNS248470271; BioRad) that hybridizes specifically to the intronic region between exons 3 and 4 of the *Il1r1* gene, which is within the floxed region of the *Il1r1*^FL allele. Efficiency of the probe was confirmed on samples isolated from *Il1r1*^FL/FL *Vav*-iCre^+/+ control mice. Whole genome amplification was confirmed on all samples using a reference HEX probe (Assay ID dMmuCNS796829161; BioRad) that hybridizes specifically to the intronic region between exons 1 and 2 of the *Il1r1* gene (non-deleted region, located upstream of the floxed region of the *Il1r1*^Lox allele). Non-template controls were included to reliably define the gating strategy. Plates were read on a QX200 droplet reader (Bio-Rad) and results analyzed using Quanta-Soft v1.5.38.1118 software (Bio-Rad). Frequency of deleted cells was calculated as "$r = (1 - 2 \times FA)/(1 - FA)$" where "r" is the frequency of deleted cells and "FA" is the fractional abundance.

### Statistical analysis

Statistical significance was determined by two-sided *t*-test, 1- or 2-way ANOVA combined with Tukey's, Sidak's or Dunnett's multiple comparisons tests, two-sided Fisher's exact test, and Kruskal–Walis test with Dunn's multiple comparisons test, as indicated in figure legends. Statistical tests were performed with GraphPad Prism software. No statistical method was used to predetermine sample size, and experiments were not randomized. The investigators were not blinded to allocation during experiments and outcome assessment.

### Reporting summary

Further information on research design is available in the Nature Portfolio Reporting Summary linked to this article.

## Data availability

The RNA-Sequencing data generated in this study have been deposited in NCBI Gene Expression Omnibus (GEO) and are accessible through GSE121249 (NCBI tracking system #19505382). The source data for Figs. 1–6 and Supplementary Figs. 1-2, 4-6 are provided as a Source Data file. All other data that supports the findings of this study are available from the corresponding authors upon request. Source data are provided with this paper.

## Code availability

Code for the RNA sequencing analysis included in this study is available at https://zenodo.org/record/8283315.

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

## Acknowledgements

We thank P. Frenette and S. Pinho (Albert Einstein College of Medicine) for help with whole mount imaging; T. Moreau and L. Mayer (University of Cambridge) for help with Nbeal2$^{-/-}$ experiments; M. Clarke (University of Cambridge) for help with IL-1 analysis. Biomedical Services at Oxford University for animal support; P. Sopp, S-A. Clark from the WIMM FACS facility (supported by the MRC HIU; MRC MHU (MC_UU_12009); NIHR Oxford BRC and John Fell Fund (131/030 and 101/517), the EPA fund (CF182 and CF170) and by the WIMM Strategic Alliance awards G0902418 and MC_UU_12025); and C. Lagerholm and D. Waithe from the Wolfson Imaging Centre Oxford (supported by MRC via the WIMM Strategic Alliance (G0902418), the MHU (MC_UU_12009), the HIU (MC_UU_12010), the Wolfson Foundation (Grant 18272), the MRC/BBSRC/EPSRC grant (MR/K015777X/1) to MICA–Nanoscopy Oxford (NanO): Novel Super-resolution Imaging Applied to Biomedical Sciences, Micron (107457/Z/15Z) and the WIMM Strategic Alliance awards G0902418 and MC_UU_12025). This work was supported by the following grants; a Kay Kendall Leukaemia Fund Junior Research Fellowship (KKL832) to T.C.L.; an EMBO-LTF (ALTF1228-2011) to T.C.L.; a Sir Henry Dale Fellowship from the Wellcome Trust and The Royal Society (210424/Z/18/Z) to T.C.L; the MRC UK (G0801073 and MC_UU_12009/5) to S.E.W.J.; the Swedish Research Council to S.E.W.J.; the Knut och Alice Wallenberg Foundation to S.E.W.J.; the Tobias Foundation to S.E.W.J.; StratRegen KI to S.E.W.J, and The Torsten Söderberg Professorial Chair in Medicine to S.E.W.J.

## Author contributions

Conceptualization: T.C.L. and S.E.W.J.; Validation: T.C.L., N.B and A.G.; Formal analysis: T.C.L., A.G., N.B. and S.E.W.J; Investigation: T.C.L., N.B., J.C., A.G., S.M., R.N., A.A., B.W., J.A.G., A.R-M., B.W., T.B.J., I.C.M and M.J.; Resources: H.N. and G.Z. for providing the NIT E antibody; M.J.R and R.D.B. for providing the Il1r1$^{FL/FL}$ mice; Data curation: T.C.L., N.B. and A.G.; Writing-Original Draft: T.C.L. and S.E.W.J.; Writing-Review & Editing: T.C.L. and S.E.W.J.; Visualization T.C.L. and N.B.; Supervision and input on experimental design and analysis: S.E.W.J., C.G., C.N. and A.J.M.; Project Administration: T.C.L. and S.E.W.J.; Funding Acquisition: T.C.L. and S.E.W.J. All authors read and approved the final version of the manuscript.

## Funding

## Competing interests

H.N. is the President and CEO, and G.Z. is an employee of CCOA Therapeutics Inc. Both authors are inventors of the monoclonal antibody NIT E (Canada patent number: 2, 689, 726; US patent number: US8, 323, 652 B2; and European patent number: 2186829). All other authors declare no competing interests.

## Additional information

[1]Haematopoietic Stem Cell Biology Laboratory, MRC Weatherall Institute of Molecular Medicine, Radcliffe Department of Medicine, University of Oxford, OX3 9DS Oxford, UK. [2]MRC Molecular Haematology Unit, MRC Weatherall Institute of Molecular Medicine, Radcliffe Department of Medicine, University of Oxford, OX3 9DS Oxford, UK. [3]Centre for Inflammatory Disease, Department of Immunology and Inflammation, Imperial College London, W12 0NN London, UK. [4]Department of Life Sciences, Imperial College London, SW7 2AZ London, UK. [5]Molecular and Cellular Immunology Section, UCL Great Ormond Street Institute of Child Health, London, UK. [6]Human Technopole, Viale Rita Levi-Montalcini 1, 20157 Milan, Italy. [7]Center for Hematology and Regenerative Medicine, Department of Medicine Huddinge, Karolinska Institutet, Karolinska University Hospital, SE-141 86 Stockholm, Sweden. [8]Department of Haematology, University of Cambridge, Cambridge, UK. [9]National Health Service (NHS) Blood and Transplant, Cambridge Biomedical Campus, Cambridge, UK. [10]Earlham Institute, Norwich Research Park, NR4 7UZ Norwich, UK. [11]Toronto Platelet Immunobiology Group and Department of Laboratory Medicine, Keenan Research Centre for Biomedical Science, St. Michael's Hospital, Toronto, ON M5B 1W8, Canada. [12]CCOA Therapeutics Inc, Toronto, ON M5B 1T8, Canada. [13]Department of Laboratory Medicine and Pathobiology, University of Toronto, Toronto, ON M5S 1A1, Canada. [14]Canadian Blood Services Centre for Innovation, Toronto, ON M5B 1W8, Canada. [15]Department of Biomedical Science, Charles E. Schmidt College of Medicine and Stiles-Nicholson Brain Institute, Florida Atlantic University, Jupiter, FL 33458, USA. [16]Division of Pharmaceutical Sciences, James L. Winkle College of Pharmacy, University of Cincinnati, Cincinnati, OH 45267, USA. [17]Department of Cell and Molecular Biology, Karolinska Institutet, SE-171 77 Stockholm, Sweden. [18]Department of Hematology, Karolinska University Hospital, Stockholm, Sweden. ✉e-mail: t.luis@imperial.ac.uk; sten.jacobsen@imm.ox.ac.uk

