## [Peer Review File · Nature Communications]

Reviewers' comments:

Reviewer #1 (Remarks to the Author):

Hematopoietic stem cells (HSCs) are regulated by the bone marrow niche. In this manuscript, using an antibody-mediated platelet depletion model, Luis et al investigated how HSCs respond to thrombocytopenia. The authors found that thrombocytopenia drives HSCs into cycle and production of platelets to restore their homeostasis. Through systematic gene expression profiling, the authors found that the bone marrow niche, particularly the perivascular LepR⁺ cells, has significant changes in the number and gene expression (e.g. IL1 pathway). IL1 pathway was pursued further functionally. Conditional IL1r1 deletion established that it is required in LepR⁺ cells but not HSCs for driving HSCs into cycle. Most conclusions are well supported by experimental data. However, the idea that depletion of a specific blood cell lineage (particularly platelets) can regulate the bone marrow niche, which in turn regulates HSC function is not a new one. This undermines the novelty of the current manuscript. Below are specific comments:

Major:

1. Although not explicitly stated in the manuscript, the antibody-mediated thrombocytopenia is essentially the immune thrombocytopenia described by several previous papers (e.g. Herd et al Blood Adv. 2021 5(23): 4877, Ramasz et al Blood. 2019 134:1046,). These papers should be cited and discussed. These papers described antibody-mediated platelet depletion activates HSCs by different mechanisms. In particular, a niche-mediated mechanism has been published (Herd et al Blood Adv. 2021 5(23): 4877). This limits the conceptual advance of the current manuscript.
2. Why there is an initial reduction of vwf⁻ HSCs in Fig 1e?
3. HSC function is defined by multilineage reconstitution. In Fig 1i, how about the lymphoid potential?
4. Quiescent HSCs are more potent in reconstituting recipient mice. But In Figs 1i and j, why less cycling HSCs are not more potent than HSCs that cycle more? Some explanation/discussion would be helpful.
5. Careful characterization of the genetic models is needed. In Fig 3i, the authors tried to rule out the possibility that HSCs directly receive IL1 signal by showing the lack of an HSC activation in IL1r1^{fl/fl}; Vav1-cre mice. A high deletion efficiency is critical for this interpretation. Similarly, in Fig S5a, the deletion efficiency is not very good. But it is not clear why the phenotype is like IL1r1^{-/-}.

6. Treating mice with NEU depletes platelets without activation, but did not lead to HSC activation and increase in number, suggesting that activation of platelets is required for HSC activation. Then is this really a response to thrombocytopenia? How about platelet activation without depletion? Does this lead to HSC activation?

Minor:

1. Line 96, typo for interferons.

2. It is not really that GPIba is undetectable on HSCs as shown in Extended Data Fig. 1a. Also the authors should show standard deviation. How about its expression in other hematopoietic cells?

3. Fig 1g is missing error bars.

4. Line 240, Fig. S5E is not in the Figures.

5. Line 334, Fig. 6j is a typo.

Reviewer #2 (Remarks to the Author):

The manuscript provides fascinating insights into the changes in the bone marrow microenvironment that lead to megakaryocyte-primed HSC expansion in response to acute platelet depletion. In a series of experiments using WT and transgenic mouse models, the authors demonstrate that treatment with anti-GPIba antibody – which is thought to activate platelets prior to clearance – leads to vWF+ HSCs entering cell cycle and a markedly quicker appearance of MKs. They found an upregulation of pro-inflammatory markers in BM niche cells following platelet depletion and in particular an IL-1 signature, suggesting a role for this cytokine in driving vWF+ HSC expansion. This was confirmed by finding higher levels of IL-1a in BM extracellular fluid that follows a similar time course of platelet depletion and vWF+ HSC activation, a mechanism that was confirmed by providing exogenous IL-1 in the absence of platelet

depletion. The rest of the manuscript focuses on identifying the roles of CBM-PV cells in sensing IL-1 released (presumably) from activated platelets. They demonstrate a clear upregulation of IL1-R specifically in CBM-PV cells, and a reversal of IL-1 regulated gene expression in mice lacking IL-1R following platelet depletion. Using a LEPR-cre specific IL-1R KO mouse model, they were able to detect a modest reduction in vWF+ HSC activation, which supports their model, but also highlights the complexity of these interactions and suggests other factors may also play key roles. Finally, the authors focus on mechanisms of platelet clearance, comparing GP1ba (platelet activation and clearance) with neuramidase treatment, which induces clearance via desialylation rather than activation. Interestingly they find that this mechanism of platelet clearance failed to activate vWF+ HSCs, and further supported this hypothesis by using a Nbeal2 knockout mouse (which has agranular platelets) failed to fully activate vWF+ after GP1ba depletion.

Overall, this is an excellent and substantial unit of work and the authors should be congratulated on covering so many aspects of HSC and platelet biology. This significantly furthers our understanding of how platelet depletion can lead to significant changes in the BM to support a rapid and sustained recovery of circulating platelets. I would recommend that the manuscript be accepted with a few suggestions for discussion listed below.

1. You elegantly show that GP1ba-mediated platelet depletion is able to activate vWF+ HSCs and that platelet activation and presumably granule release is essential to this. You have also shown that desialylation that clears platelets without significant activation fails to replicate this response. However, as ITP is most commonly caused by antibodies to other platelet glycoproteins (primarily GPIIb/IIIa and GPIbIX) that lead to platelet clearance by opsonization rather than activation, could you possibly discuss these potential mechanisms in the discussion section?
2. You mention that TPO levels increase in response to platelet clearance – presumably due to reduced platelet sponging of circulating TPO – and that this may also influence vWF+ HSC activation. Did you determine whether the levels of TPO had increase in the BM extracellular fluid?
3. A very minor point – you refer the “star methods” in Fig1 legend. I don’t think this is applicable to this journal.

Reviewer #3 (Remarks to the Author):

The manuscript by Luis et al describes the specific cell cycle activation of the megakaryocytic/platelet biased VWF+ HSC subset, previously described by the same lab, 1 day upon platelets depletion via anti-GPIba antibody treatment. The authors claim this effect requires platelet activation and is mediated via IL-1 activation of IL1R+ LeptinR+ cells.

The manuscript is well written and it's clear that the authors performed an extensive amount of testing, including extensive RNAseq and cell cycle analyses of rare subsets of stroma and HSCs to explain their results. However, overall, I have several major concerns regarding data interpretation regarding the specific role of platelets in VWF+ HSCs activation and my enthusiasm for the study drops as the authors failed to elaborate how exactly IL-1 impacts LeptinR+ -mediated activation of VWF+ HSCs.

Specific comments:

1. My first major concern is that it's still unclear if mechanistically the effect of anti-GPIIb/IIIa antibody on VWF+ HSCs activation is due to platelet specific deletion or via alterations in the megakaryocytic niche. Previous studies already demonstrated the role of megakaryocytes and their secreted factors on the regulation of HSC quiescence/proliferation (PMID:25326802; PMID:25326798; PMID:25451253), in particular the VWF+ HSC subpopulation (PMID:29456137). The observed results can simply be explained by alterations on the megakaryocytic lineage or their secreted niche factors such as PF4 upon treatment.

The effect of anti-GPIIb/IIIa antibody on MKPs and more mature MKs is still unclear with opposite reports in the literature. Although the authors (lines 141-144) claim that MKPs and MKs were not depleted following antibody administration, their own results Ext Data Fig 1d show a clear reduction in the number of MkP, 1 day following antibody administration.

More concerning is the authors negative result with Neuraminidase treatment (Fig. 6) which is known to deplete platelets without affecting megakaryocytes. Although the authors claim that it is the lack of platelets activation in Neuraminidase treatment in contrast to anti-GPIIb/IIIa, this result corroborates my main concern. A simple explanation for the observed effect is the previously reported role of megakaryocytes in the regulation of VWF+ HSCs.

It's also missing to evaluate the effect of anti-GPIIb/IIIa treatment on the levels of PF4, TGF β , Fgf1 and TPO.

2. My second major concern is that although Il1r1-null mice and LepR-Cre, Il1r1-null mice show a significant reduction in the activation of Vwf+ HSCs upon anti-GPIIb/IIIa treatment (Figure 3e and 5a), this effect is still very small which suggests that IL-1/IL-1R signal is a minor mechanism in HSCs activation upon anti-GPIIb/IIIa treatment.

Accordingly, line 230-234: Fig 3d clearly shows that both Vwf+ and Vwf- HSC subsets respond to IL1, it's unclear the differential results that the authors try to establish between these HSC subsets and the parallel with anti- GPIIb/IIIa treatment. If there are differences, they are minor and not clear by the authors choice of graphs.

Figure 3i - considering the previous literature on HSCs direct response to IL-1, it would be important to confirm in in vitro cultures the non-activation of Vwf+ and Vwf- HSCs in response to direct IL-1 exposure. How do the authors explain this discrepancy with previous studies? Also, does Vav-Cre mice recombine with the same efficiency in both Vwf+ and Vwf- HSC subsets?

3. Finally, the authors failed to clarify the mechanism by which IL-1 impacts LeptinR+ -mediated activation of VWF+ HSCs.

Perivascular niche cells sense thrombocytopenia and activate hematopoietic stem cells in an IL-1-dependent manner

NCOMMS-22-12571-T

Point-by-point response to reviewers' comments:

Reviewer #1 (Remarks to the Author):

Hematopoietic stem cells (HSCs) are regulated by the bone marrow niche. In this manuscript, using an antibody-mediated platelet depletion model, Luis et al investigated how HSCs respond to thrombocytopenia. The authors found that thrombocytopenia drives HSCs into cycle and production of platelets to restore their homeostasis. Through systematic gene expression profiling, the authors found that the bone marrow niche, particularly the perivascular LepR⁺ cells, has significant changes in the number and gene expression (e.g. IL1 pathway). IL1 pathway was pursued further functionally. Conditional IL1r1 deletion established that it is required in LepR⁺ cells but not HSCs for driving HSCs into cycle. Most conclusions are well supported by experimental data. However, the idea that depletion of a specific blood cell lineage (particularly platelets) can regulate the bone marrow niche, which in turn regulates HSC function is not a new one. This undermines the novelty of the current manuscript. Below are specific comments:

Response: We appreciate that this reviewer finds that “most conclusions are well supported by the data”. We also thank the reviewer for raising some critical questions regarding the novelty of the reported findings, which we have addressed in our point-by-point responses below, and also clarified further in the revised manuscript. While we certainly agree that “the idea that depletion of a specific blood cell lineage (particularly platelets) can regulate the bone marrow niche, which in turn regulates HSC function is not a new one”, data from previous studies referenced have not demonstrated this with critical genetic experiments as outlined in more detail in response to this particular issue (point 1). We have also addressed the other issues raised, in part by including new data as specified below.

Major:

1. Although not explicitly stated in the manuscript, the antibody-mediated thrombocytopenia is essentially the immune thrombocytopenia described by several previous papers (e.g. Herd et al Blood Adv. 2021 5(23): 4877, Ramasz et al Blood. 2019 134:1046,). These papers should be cited and discussed. These papers described antibody-mediated platelet depletion activates HSCs by different mechanisms. In particular, a niche-mediated mechanism has been published (Herd et al Blood Adv. 2021 5(23): 4877). This limits the conceptual advance of the current manuscript.

We do indeed agree that the conceptual advance of our manuscript would be compromised if “a niche-mediated mechanism has been published” when it comes to demonstrating that loss of a mature blood cell lineage (in this case platelets) results in activation of multipotent HSCs through a specific bone marrow niche cell population strongly implicated in regulation of HSCs.

While these 2 previous publications (Herd et al., 2021; Ramasz et al., 2019) do suggest that HSC activation is associated with changes in the bone marrow microenvironment, they fail to provide any direct evidence for a specific role of the bone marrow microenvironment or HSC niches, mainly due to the lack of experimental evidence where one of the identified signals or signalling pathways (which also are active in multiple other tissues) is knocked-out in a specific bone marrow niche cell population, unlike in our studies. Thus, the novelty of our main finding is not compromised, as our studies constitute the first demonstration that one specific signal (IL-1) acting on one specific niche cell population (LepR⁺ perivascular cells) is critical for linking the consumption of a mature blood cell lineage (platelets) and activation of HSCs.

Moreover, Herd et al (Blood Advances, 2021), used platelet depletion achieved by repeated injection of an antibody against CD41, which has been demonstrated to also be expressed on HSCs and is

critical for their function (Boisset et al., 2013; Gekas and Graf, 2013), precluding any definitive conclusion on the role of the niche in this process. Moreover, the link between HSC activation and long-term niche changes suggested by Herd et al is purely correlative, not supported by any loss- or gain-of- function studies targeted to any specific niche or bone marrow stroma cells.

Ramasz et al (Blood, 2019), implicate the involvement of 3 different signalling pathways in this process but did not provide any specific experimental evidence that these signals derive from the bone marrow microenvironment or niches. This because the only cell specific loss-of-function studies were performed by depleting megakaryocytes (and not by knocking-out the implicated signalling pathways in megakaryocytes), which as pointed out also by reviewer 3 has been shown to regulate HSCs already in homeostasis, but through different mechanisms (Bruns et al., 2014; Nakamura-Ishizu et al., 2014; Pinho et al., 2018; Zhao et al., 2014). Other functional analyses were performed by administration of blocking antibodies or inhibitors, which also inactivate these signals in other tissues in a systemic way and therefore the specific role of the bone marrow niche could not be demonstrated. We have however the revised manuscript referenced and discussed these papers in light of our findings, although as outlined we don't find that they compromise the novelty of our manuscript (Page 5, line 193). We have also clarified that GPIIb/IIIa antibody administration is a model of Immune Thrombocytopenia Purpura (Page 3, line 119).

2. Why there is an initial reduction of vwf- HSCs in Fig 1e?

The initial reduction of Vwf- HSCs could be explained by either Vwf- HSCs differentiating to generate downstream progenitors and/or Vwf- HSCs up-regulating Vwf it-self. Distinguishing between these (and other) possibilities would require complex fate-mapping analysis with a transgenic mouse where Cre is under control a gene promoter exclusively active in either Vwf- or Vwf+ HSCs and not in downstream blood cell populations. Such a gene is yet to be identified making it difficult to answer this question. Nevertheless, we have addressed this issue in our revised discussion (Page 10, line 389), and most importantly this finding is not important to the interpretation of the main conclusions of our manuscript.

3. HSC function is defined by multilineage reconstitution. In Fig 1i, how about the lymphoid potential?

*Both HSC subsets efficiently repopulate also the lymphoid lineages and these data have now been included in the revised manuscript (Revised Fig1i; text page pages 4-5, line 161). The decision on showing only the myeloid and platelet lineages was based on the fact that mature lymphoid cells are long lived, and the myeloid/platelet cell lineages are the ones with the highest turnover and therefore considered to more reliably provide a more accurate representation of HSC reconstitution (Carrelha et al., 2018). Moreover, to further demonstrate that multilineage long-term HSCs are activated after platelet depletion we additionally used the doxycycline-inducible (tet-ON) H2B-mCherry mice. In line with the Biotin model, FACS sorted mCherry^{lo} (proliferative) LSKFlt3⁻CD48⁻CD150⁺ cells sorted from platelet-depleted mice (Extended Data Fig.2i and pages 4-5, line 161) had *in vivo* long-term (16 weeks) multilineage reconstitution potential.*

4. Quiescent HSCs are more potent in reconstituting recipient mice. But In Figs 1i and j, why less cycling HSCs are not more potent than HSCs that cycle more? Some explanation/discussion would be helpful.

Biotin labelling allows the fate-mapping of cells with a history of proliferation so that lower-biotin labelling does not mean that the cells are still proliferating. In fact, in these experiments, HSCs were transplanted 2 days post platelet depletion, when the majority of HSCs are no longer actively proliferating (Fig. 1c-d). These experiments were designed in this way to precisely avoid the potentially confounding effect of cell cycle in reconstitution capacity, as indicated by this reviewer. The same applies to our new data using the doxycycline-inducible (tet-ON) H2B-mCherry system (Extended Data Fig.2i). On a related note, the "evidence" linking HSC quiescence with increased reconstitution potential remains purely

correlative and may alternatively reflect different purity of phenotypically defined quiescent and cycling HSCs.

5. Careful characterization of the genetic models is needed. In Fig 3i, the authors tried to rule out the possibility that HSCs directly receive IL1 signal by showing the lack of an HSC activation in IL1r1^{fl/fl}; Vav1-cre mice. A high deletion efficiency is critical for this interpretation. Similarly, in Fig S5a, the deletion efficiency is not very good. But it is not clear why the phenotype is like Il1r1^{-/-}.

Given the very low/undetectable levels of Il1r1 mRNA in HSCs that we (Figure 3g-h) and others observed (Kovtonyuk et al., 2022; Pietras et al., 2016), the analysis of Il1r1 deletion in HSCs requires analysis of genomic DNA. To address this we have now included new digital-drop PCR data on Vwf⁺ and Vwf⁻ HSCs isolated from Il1r1^{FL/FL}Vav-Cre^{Tg/+} mice (Page 20, line 820; methods) This new data demonstrates virtually complete and similar deletion efficiency in both HSC subsets (Extended Data Fig.4g; text Page 7 line 265, and page 10 line 406). Vav-iCre is a well-established and widely used mouse line, known to achieve very high recombination efficiency of many different targeted genes (de Boer et al., 2003; Ding and Morrison, 2013; Luis et al., 2016; Nomura et al., 2021) and the deletion efficiency we observed here is in line with those previous studies. Moreover, in Wt mice, platelet depletion leads to up to 70% of HSCs entering cell cycle (Fig.1c-d) and therefore, the absence of HSCs recruitment into proliferation after platelet depletion in Vav1-Cre:Il1r1^{FL/FL} mice (Fig.3i) could not be explained by a minority of HSCs escaping efficient recombination in this model.

The fact that germline and Lepr-Cre-induced Il1r1 deficiency achieve the same HSC proliferation impairment in response to platelet depletion despite the approximate 50% deletion efficiency in the later is indeed somewhat surprising. This should however not distract from the key finding that we observe a significant reduction in HSC activation in response to platelet depletion when specifically deleting Il1r1 in LepR⁺ cells. Importantly, other studies have also shown strong biological effects of Lepr-Cre induced deletion of signalling pathways (Ding et al., 2012), although the deletion efficiency is not very high. This is compatible with Lepr-Cre recombining efficiently in the relevant cells.

6. Treating mice with NEU depletes platelets without activation, but did not lead to HSC activation and increase in number, suggesting that activation of platelets is required for HSC activation. Then is this really a response to thrombocytopenia? How about platelet activation without depletion? Does this lead to HSC activation?

The mechanism we propose here shows that HSC proliferation is triggered by platelet activation, which subsequently leads to thrombocytopenia. Activated platelets are rapidly and efficiently targeted for degradation in the spleen and liver as a normal process (Quach et al., 2018). However, we agree with this reviewer that investigating the effect of platelet activation in the absence of depletion would be interesting, but given the nature of the process it is according to the platelet experts involved in this manuscript (Dr Cedric Ghevaert and Prof Heyu Ni) not possible to perform as platelet activation and subsequent clearance cannot be uncoupled. In our revised manuscript we further discuss the importance of platelet activation for the induction of HSC proliferation (page 11 line 446, and page 10 line 379).

Minor:

1. Line 96, typo for interferons. *We thank the reviewer for highlighting this. It has been corrected.*

2. It is not really that GPIIb is undetectable on HSCs as shown in Extended Data Fig. 1a. Also the authors should show standard deviation. How about its expression in other hematopoietic cells?

We apologize for any misunderstanding this figure may have created. In Extended Data Fig1a GPIIb staining is denoted in red and Isotype control in grey. We have provided new data and standard deviation for all mice, more clearly showing that there is no detectable expression of GPIIb/Cd42b on HSCs. Our findings are further supported by other previous studies also showing no expression of this

gene in HSC enriched cell populations (Gekas and Graf, 2013). Data for other haematopoietic populations is provided in the same figure.

3. Fig 1g is missing error bars.

This graph shows percentage of cells and therefore does not have error bars. Statistical analysis was performed by Fisher's exact test, the recommended statistical method for this type of data sets.

4. Line 240, Fig. S5E is not in the Figures. *We thank the reviewer for highlighting this. It has now been corrected.*

5. Line 334, Fig. 6j is a typo. *We thank the reviewer for highlighting this. It has now been corrected.*

Reviewer #2 (Remarks to the Author):

The manuscript provides fascinating insights into the changes in the bone marrow microenvironment that lead to megakaryocyte-primed HSC expansion in response to acute platelet depletion. In a series of experiments using WT and transgenic mouse models, the authors demonstrate that treatment with anti-GPIIb antibody – which is thought to activate platelets prior to clearance – leads to vWF+ HSCs entering cell cycle and a markedly quicker appearance of MKs. They found an upregulation of pro-inflammatory markers in BM niche cells following platelet depletion and in particular an IL-1 signature, suggesting a role for this cytokine in driving vWF+ HSC expansion. This was confirmed by finding higher levels of IL-1a in BM extracellular fluid that follows a similar time course of platelet depletion and vWF+ HSC activation, a mechanism that was confirmed by providing exogenous IL-1 in the absence of platelet depletion. The rest of the manuscript focuses on identifying the roles of CBM-PV cells in sensing IL-1 released (presumably) from activated platelets. They demonstrate a clear upregulation of IL1-R specifically in CBM-PV cells, and a reversal of IL-1 regulated gene expression in mice lacking IL-1R following platelet depletion. Using a LEPR-cre specific IL-1R KO mouse model, they were able to detect a modest reduction in vWF+ HSC activation, which supports their model, but also highlights the complexity of these interactions and suggests other factors may also play key roles. Finally, the authors focus on mechanisms of platelet clearance, comparing GP1ba (platelet activation and clearance) with neuramidase treatment, which induces clearance via desialylation rather than activation. Interestingly they find that this mechanism of platelet clearance failed to activate vWF+ HSCs, and further supported this hypothesis by using a Nbeal2 knockout mouse (which has agranular platelets) failed to fully activate vWF+ after GP1ba depletion. Overall, this is an excellent and substantial unit of work and the authors should be congratulated on covering so many aspects of HSC and platelet biology. This significantly furthers our understanding of how platelet depletion can lead to significant changes in the BM to support a rapid and sustained recovery of circulating platelets. I would recommend that the manuscript be accepted with a few suggestions for discussion listed below.

Response: We highly appreciate the very positive feedback from this reviewer and for nicely summarizing the key and novel findings from our manuscript. We have addressed the specific points raised as outlined below.

1. You elegantly show that GP1ba-mediated platelet depletion is able to activate vWF+ HSCs and that platelet activation and presumably granule release is essential to this. You have also shown that desialylation that clears platelets without significant activation fails to replicate this response. However, as ITP is most commonly caused by antibodies to other platelet glycoproteins (primarily GPIIb/IIIa and GPIbIX) that lead to platelet clearance by opsonization rather than activation, could you possibly discuss these potential mechanisms in the discussion section?

As suggested, we discussed potential effects of other anti-platelets antibodies found in ITP patients, particularly the ones that lead to platelet opsonization (without activation) and degradation phagocyte-mediated depletion (Page 10, line 379).

2. You mention that TPO levels increase in response to platelet clearance – presumably due to reduced

platelet sponging of circulating TPO – and that this may also influence vWF+ HSC activation. Did you determine whether the levels of TPO had increase in the BM extracellular fluid?

TPO levels post platelet depletion have been measured systemically (blood serum) in our previous study (Sanjuan-Pla et al., 2013) but not in bone marrow serum. In this study we showed that TPO levels indeed inversely correlate with platelet count, and as briefly discussed on pages 7 (line 250) and 11 (line 426) we believe systemic TPO in combination with IL-1 might synergistically regulate the full response of HSCs to platelet depletion, as previous studies have shown that TPO regulates HSCs systemically and is almost exclusively produced in the liver (Decker et al., 2018).

3. A very minor point – you refer the “star methods” in Fig1 legend. I don’t this this is applicable to this journal.

We thank the reviewer for highlighting this. It has now been corrected.

Reviewer #3 (Remarks to the Author):

The manuscript by Luis et al describes the specific cell cycle activation of the megakaryocytic/platelet biased VWF+ HSC subset, previously described by the same lab, 1 day upon platelets depletion via anti-GPIIb antibody treatment. The authors claim this effect requires platelet activation and is mediated via IL-1 activation of IL1R+ LeptinR+ cells.

The manuscript is well written and it’s clear that the authors performed an extensive amount of testing, including extensive RNAseq and cell cycle analyses of rare subsets of stroma and HSCs to explain their results. However, overall, I have several major concerns regarding data interpretation regarding the specific role of platelets in VWF+ HSCs activation and my enthusiasm for the study drops as the authors failed to elaborate how exactly IL-1 impacts LeptinR+ -mediated activation of VWF+ HSCs.

Specific comments:

1. My first major concern is that it’s still unclear if mechanistically the effect of anti-GPIIb antibody on vWF+ HSCs activation is due to platelet specific deletion or via alterations in the megakaryocytic niche. Previous studies already demonstrated the role of megakaryocytes and their secreted factors on the regulation of HSC quiescence/proliferation (PMID:25326802; PMID:25326798; PMID:25451253), in particular the VWF+ HSC subpopulation (PMID:29456137). The observed results can simply be explained by alterations on the megakaryocytic lineage or their secreted niche factors such as PF4 upon treatment.

As outlined in more detail below to the specific points regarding this, we agree that a role also of megakaryocytes cannot be ruled out (as emphasized in the revised manuscript on pages 11-12, line 446), but this should not distract from our results specifically demonstrating a role of platelets (and their activation) in the observed HSC response to the anti-GPIIb treatment.

The effect of anti-GPIIb antibody on MKPs and more mature MKs is still unclear with opposite reports in the literature. Although the authors (lines 141-144) claim that MKPs and MKs were not depleted following antibody administration, their own results Ext Data Fig 1d show a clear reduction in the number of MkP, 1 day following antibody administration.

It is true that there is some reduction (Ext Fig1d) although not significant ($p=0.82$) in MkPs at 1 day post platelet depletion with GPIIb antibody. This has been clarified in the revised manuscript (Page 4, line 143). This reduction may potentially reflect the differentiation of MkPs to generate downstream mature MKs which are in fact slightly increased (Ext Fig 1g), necessary to replenish platelet numbers. Nevertheless, we agree that there is a likely role also of Mks in the observed HSC activation, but this does not affect our results demonstrating a role of platelets (and their activation) in the observed HSC response to the anti-GPIIb treatment (see points below).

More concerning is the authors negative result with Neuraminidase treatment (Fig. 6) which is known

to deplete platelets without affecting megakaryocytes. Although the authors claim that it is the lack of platelets activation in Neuraminidase treatment in contrast to anti-GPIIb/IIIa, this result corroborates my main concern. A simple explanation for the observed effect is the previously reported role of megakaryocytes in the regulation of VWF+ HSCs.

We agree that the lack of HSC proliferation in Neuraminidase-treated mice is compatible with a role also for MKs in this process (Fig. 6e), something we point out on pages 9 (line 351) and 11-12 (line 446) of the revised manuscript. Therefore, the experiments (summarized in Fig. 6h) where GPIIb/IIIa antibody was administered following Neuraminidase treatment, were key for a more definitive conclusion that platelets themselves play a role in the observed activation. Importantly, through this experimental approach, by the time of GPIIb/IIIa antibody administration platelets have already been depleted completely whereas MKs are still present. The fact that the sequential NEU-GPIIb/IIIa treatment resulted in significantly reduced cell cycle activation of HSCs, when compared to GPIIb/IIIa treatment alone, therefore established the involvement of platelets in this process (Fig. 6h). We have made this important point more clear in the revised discussion (Pages 11-12, line 446).

It's also missing to evaluate the effect of anti-GPIIb/IIIa treatment on the levels of PF4, TGF β , Fgf1 and TPO.

To gain insight into the contribution of other potential HSC regulators we have now included transcriptional data of Pf4, Tgfb and Fgf1 expression in central bone marrow LepR+ perivascular and endothelial cells, and in megakaryocytes (Extended Data Fig 4d; text page 7, line 250; and page 11, line 426). These data show increased expression of Pf4 in central bone marrow endothelial cells. The expression of Tgfb and Fgf1 by the different niche cells analyzed was not different in homeostasis and after platelet depletion. Physiologically relevant Thpo is almost exclusively produced in the distant liver (Decker et al., Science, 2018) and we have previously shown that the levels of this cytokine in serum is also increased following platelet depletion (SanJuan-Pla, et al, Nature, 2013). Together this suggests that both Pf4 and Thpo may synergistically operate with IL-1 to promote the optimal activation of HSC following platelet depletion, something we point out in the revised manuscript on pages 7 (line 250) and 11 (line 426).

While interesting to explore the effects of other factors previously implicated in HSC regulation, this should not distract from the fact that we performed a line of genetic experiments that for the first time establish that one specific signal (IL-1) acting on one specific niche cell population (LepR+ perivascular cells) is critical for linking the consumption of a mature blood cell lineage (platelets) and activation of HSCs, and as specifically pointed out by Reviewer 2 already represents a "substantial unit of work"... "covering so many aspects of HSC and platelet biology."

2. My second major concern is that although Il1r1-null mice and LepR-Cre, Il1r1-null mice show a significant reduction in the activation of Vwf+ HSCs upon anti-GPIIb/IIIa treatment (Figure 3e and 5a), this effect is still very small which suggests that IL-1/IL-1R signal is a minor mechanism in HSCs activation upon anti-GPIIb/IIIa treatment.

We agree and point out in the revised manuscript (page 7, line 250; and page 11, line 426) that our findings in addition to demonstrating a role of IL-1 targeting LepR+ cells suggest that other pathways are involved in the HSC activation in response to platelet activation/depletion. That multiple pathways are involved in complex feedback mechanisms as this is not so surprising, and therefore expectable that interfering with one pathway alone will not fully abrogate the optimal HSC response. Nevertheless, we in our manuscript conclusively establish a role of IL-1 signalling through LepR+ cells, with the genetic experiments performed, as this reviewer points out by establishing that "Il1r1-null mice and LepR-Cre, Il1r1-null mice show a significant reduction in the activation of Vwf+ HSCs upon anti-GPIIb/IIIa treatment (Figure 3e and 5a)". Reviewer 2 also emphasizes (like this reviewer) that our studies also support "the complexity of these interactions and suggests other factors may also play key roles", while at the same time s/he enthusiastically recommends publication and points out that "this significantly furthers our

understanding of how platelet depletion can lead to significant changes in the BM to support a rapid and sustained recovery of circulating platelets.”

Accordingly, line 230-234: Fig 3d clearly shows that both Vwf+ and Vwf- HSC subsets respond to IL1, it's unclear the differential results that the authors try to establish between these HSC subsets and the parallel with anti- GPIIb treatment. If there are differences, they are minor and not clear by the authors choice of graphs.

To further clarify the differential effect of IL-1 administration on Vwf+ and Vwf- HSC subsets we have now further analysed the data that established that there is a significantly higher recruitment of Vwf+ HSCs into cell cycle, when comparing with Vwf- HSCs, after administration of IL-1b (p=0.005 for 50ug; p=0.048 for 25ug IL-1b). These data are in agreement with the preferential recruitment of Vwf+ HSCs after platelet depletion with GPIIb antibody (Fig.1d) and with a role for IL-1 in this process. The new data analysis is presented in Extended Data Fig.4c and we clarified this further this on page 6 (line 243) of the revised manuscript.

Figure 3i – considering the previous literature on HSCs direct response to IL-1, it would be important to confirm in in vitro cultures the non-activation of Vwf+ and Vwf- HSCs in response to direct IL-1 exposure. How do the authors explain this discrepancy with previous studies? Also, does Vav-Cre mice recombine with the same efficiency in both Vwf+ and Vwf- HSC subsets?

The direct effect of IL-1 on HSCs was proposed based on in vitro experiments in which HSCs were FACS purified and cultured for 3 days (Pietras et al., 2016). The effect of IL-1 was only observed after 48hrs in culture and since FACS purified HSCs have been shown to easily and rapidly lose their HSC potential in vitro, they may have differentiated and no longer be true HSCs, and regardless the HSC potential of cells responding to IL-1 was not demonstrated by functional validation in those studies. The fact that HSCs might have differentiated is in line with the increased expression of IL-1R we observed in the immediate progenitors downstream of HSCs (Fig.3h). For the same reasons we do not feel that further in vitro studies are appropriate. Rather, the key experiment was the already performed Vav-Cre-mediated deletion of Il1r1 in vivo and therefore we agree that it was important to demonstrate (as shown in the revised Ext Data Fig.4g) that this resulted in complete deletion of IL-1R in Vwf+ as well as Vwf- HSCs. Given the very low/undetectable levels of Il1r1 mRNA in HSCs that we (Figure 3g-h) and others observed (Kovtonyuk et al., 2022; Pietras et al., 2016), the analysis of Il1r1 deletion in HSCs required analysis of genomic DNA. These new data demonstrate virtually complete and similar deletion efficiency in both HSC subsets (Extended Data Fig.4g and text Page 7, line 265). Moreover, in Wt mice, platelet depletion leads to up to 70% of HSCs entering cell cycle (Fig.1c-d) and therefore, the absence of HSCs recruitment into proliferation after platelet depletion in Vav1-Cre:Il1r1^{F/F} mice (Fig.3i) could not be explained by a minority of HSCs escaping efficient recombination in this model.

3. Finally, the authors failed to clarify the mechanism by which IL-1 impacts LeptinR+ -mediated activation of VWF+ HSCs.

We agree this is an interesting question (for which we provide some data, Fig.5d-g) that however does not affect and should not distract from the novelty of the main conclusions we provide, as our studies constitute the first demonstration that one specific signal (IL-1) acting on one specific niche cell population (LepR+ perivascular cells) is critical for linking the consumption of a mature blood cell lineage (platelets) and activation of HSCs. Nevertheless, in the revised manuscript we show more RNA seq data from purified HSCs to further shed light on the potential role of TGFb in this process (Fig.5g and page 8, line 317). Together, our data suggests the observed cell cycle activation of HSCs results from a shift in Smad signalling in these cells, leading to the inhibition of quiescence reinforcing TGFβ signals and the activation of BMP/Activin signalling, which was previously implicated in promoting Mk-lineage differentiation. Reviewer 2 is also in line with this stating that while the studies “suggests other factors may also play key roles”...“this is an excellent and substantial unit of work and the authors should be

congratulated on covering so many aspects of HSC and platelet biology. This significantly furthers our understanding of how platelet depletion can lead to significant changes in the BM to support a rapid and sustained recovery of circulating platelets”.

References:

- Boisset, J.C., Clapes, T., Van Der Linden, R., Dzierzak, E., and Robin, C. (2013). Integrin alphaIIb (CD41) plays a role in the maintenance of hematopoietic stem cell activity in the mouse embryonic aorta. *Biol Open* 2, 525-532. 10.1242/bio.20133715.
- Bruns, I., Lucas, D., Pinho, S., Ahmed, J., Lambert, M.P., Kunisaki, Y., Scheiermann, C., Schiff, L., Poncz, M., Bergman, A., and Frenette, P.S. (2014). Megakaryocytes regulate hematopoietic stem cell quiescence through CXCL4 secretion. *Nat Med* 20, 1315-1320. 10.1038/nm.3707.
- Carrelha, J., Meng, Y., Kettle, L.M., Luis, T.C., Norfo, R., Alcolea, V., Boukarabila, H., Grasso, F., Gambardella, A., Grover, A., et al. (2018). Hierarchically related lineage-restricted fates of multipotent haematopoietic stem cells. *Nature* 554, 106-111. 10.1038/nature25455.
- de Boer, J., Williams, A., Skavdis, G., Harker, N., Coles, M., Tolaini, M., Norton, T., Williams, K., Roderick, K., Potocnik, A.J., and Kioussis, D. (2003). Transgenic mice with hematopoietic and lymphoid specific expression of Cre. *Eur J Immunol* 33, 314-325. 10.1002/immu.200310005.
- Decker, M., Leslie, J., Liu, Q., and Ding, L. (2018). Hepatic thrombopoietin is required for bone marrow hematopoietic stem cell maintenance. *Science* 360, 106-110. 10.1126/science.aap8861.
- Ding, L., and Morrison, S.J. (2013). Haematopoietic stem cells and early lymphoid progenitors occupy distinct bone marrow niches. *Nature* 495, 231-235. nature11885 [pii] 10.1038/nature11885.
- Ding, L., Saunders, T.L., Enikolopov, G., and Morrison, S.J. (2012). Endothelial and perivascular cells maintain haematopoietic stem cells. *Nature* 481, 457-462. nature10783 [pii] 10.1038/nature10783.
- Gekas, C., and Graf, T. (2013). CD41 expression marks myeloid-biased adult hematopoietic stem cells and increases with age. *Blood* 121, 4463-4472. 10.1182/blood-2012-09-457929.
- Herd, O.J., Rani, G.F., Hewitson, J.P., Hogg, K., Stone, A.P., Cooper, N., Kent, D.G., Genever, P.G., and Hitchcock, I.S. (2021). Bone marrow remodeling supports hematopoiesis in response to immune thrombocytopenia progression in mice. *Blood Adv* 5, 4877-4889. 10.1182/bloodadvances.2020003887.
- Kovtonyuk, L.V., Caiado, F., Garcia-Martin, S., Manz, E.M., Helbling, P., Takizawa, H., Boettcher, S., Al-Shahrour, F., Nombela-Arrieta, C., Slack, E., and Manz, M.G. (2022). IL-1 mediates microbiome-induced inflammaging of hematopoietic stem cells in mice. *Blood* 139, 44-58. 10.1182/blood.2021011570.
- Luis, T.C., Luc, S., Mizukami, T., Boukarabila, H., Thongjuea, S., Woll, P.S., Azzoni, E., Giustacchini, A., Lutteropp, M., Bouriez-Jones, T., et al. (2016). Initial seeding of the embryonic thymus by immune-restricted lympho-myeloid progenitors. *Nat Immunol*. 10.1038/ni.3576.
- Nakamura-Ishizu, A., Takubo, K., Fujioka, M., and Suda, T. (2014). Megakaryocytes are essential for HSC quiescence through the production of thrombopoietin. *Biochem Biophys Res Commun* 454, 353-357. 10.1016/j.bbrc.2014.10.095.
- Nishimura, S., Nagasaki, M., Kunishima, S., Sawaguchi, A., Sakata, A., Sakaguchi, H., Ohmori, T., Manabe, I., Italiano, J.E., Jr., Ryu, T., et al. (2015). IL-1alpha induces thrombopoiesis through megakaryocyte rupture in response to acute platelet needs. *J Cell Biol* 209, 453-466. 10.1083/jcb.201410052.
- Nomura, N., Ito, C., Ooshio, T., Tadokoro, Y., Kohno, S., Ueno, M., Kobayashi, M., Kasahara, A., Takase, Y., Kurayoshi, K., et al. (2021). Essential role of autophagy in protecting neonatal haematopoietic stem cells from oxidative stress in a p62-independent manner. *Sci Rep* 11, 1666. 10.1038/s41598-021-81076-z.
- Pietras, E.M., Mirantes-Barbeito, C., Fong, S., Loeffler, D., Kovtonyuk, L.V., Zhang, S., Lakshminarasimhan, R., Chin, C.P., Techner, J.M., Will, B., et al. (2016). Chronic interleukin-1 exposure drives haematopoietic stem cells towards precocious myeloid differentiation at the expense of self-renewal. *Nat Cell Biol* 18, 607-618. 10.1038/ncb3346.
- Pinho, S., Marchand, T., Yang, E., Wei, Q., Nerlov, C., and Frenette, P.S. (2018). Lineage-Biased Hematopoietic Stem Cells Are Regulated by Distinct Niches. *Dev Cell* 44, 634-641 e634. 10.1016/j.devcel.2018.01.016.
- Quach, M.E., Chen, W., and Li, R. (2018). Mechanisms of platelet clearance and translation to improve platelet storage. *Blood* 131, 1512-1521. 10.1182/blood-2017-08-743229.

- Ramasz, B., Kruger, A., Reinhardt, J., Sinha, A., Gerlach, M., Gerbaulet, A., Reinhardt, S., Dahl, A., Chavakis, T., Wielockx, B., and Grinenko, T. (2019). Hematopoietic stem cell response to acute thrombocytopenia requires signaling through distinct receptor tyrosine kinases. *Blood* 134, 1046-1058. 10.1182/blood.2019000721.
- Sanjuan-Pla, A., Macaulay, I.C., Jensen, C.T., Woll, P.S., Luis, T.C., Mead, A., Moore, S., Carella, C., Matsuoka, S., Bouriez Jones, T., et al. (2013). Platelet-biased stem cells reside at the apex of the haematopoietic stem-cell hierarchy. *Nature* 502, 232-236. 10.1038/nature12495.
- Zhao, M., Perry, J.M., Marshall, H., Venkatraman, A., Qian, P., He, X.C., Ahamed, J., and Li, L. (2014). Megakaryocytes maintain homeostatic quiescence and promote post-injury regeneration of hematopoietic stem cells. *Nat Med* 20, 1321-1326. 10.1038/nm.3706.

REVIEWER COMMENTS

Reviewer #1 (Remarks to the Author):

The revision addressed many of the initial concerns. However, the main concerns still remain. The conceptual advance is limited given what is known.

1. Regarding the conceptual novelty of the manuscript, I appreciate that the current manuscript used rigorous mouse genetics to conditionally delete IL1R from bone marrow niche cells. However, both Herd et al and Ramasz et al provided evidence that platelet depletion can regulate HSCs by regulating the bone marrow niche. Herd et al specifically focused on LepR⁺ cells. Ramasz et al demonstrate that the HSC response to acute thrombocytopenia is mediated by SCF and Vegf from the niche. In addition, the phenotypes described by the authors in the IL1R conditional knockout mice are mild (not sure if the mice have a strong platelet production defect after depletion), suggesting this may not be the major pathway mediating HSC activation. Considering all of these points, I am not convinced that the current manuscript represents a significant conceptual advance.

2. The manuscript still needs more careful editing as there are still typos/incorrect citations that make the manuscript hard to understand. For example, Line 166, Ex Data Fig 1h showing H2B-cherry data do not exist. Line 241, it is not clear why IL1a expression by MKs are in line with reports that IL1 is mainly produced by T cells and granulocytes. Line 318, it should be 4g, not 5g.

3. Data in several places need appropriate controls or more information to support the conclusions. For example, IgG control is needed for Extended Data Fig 1e. T cell reconstitution is needed for Fig. 1I and Extended Data Fig 2i. Control Wt + IgG is needed for Fig. 3e. How were Mks purified for RNA-seq in Extended Data Fig 4b? Mks are technically challenging to sort using flow cytometry.

Reviewer #2 (Remarks to the Author):

I was supportive of this manuscript previously and I feel it has only been strengthened by the changes the authors have made. An interesting and robustly-conducted study.

Reviewer #3 (Remarks to the Author):

In response to the reviewer's comments, the authors modified the manuscript and addressed some of the original concerns. However, one of the key issues it's still not addressed. It's unclear if the observed effects are in fact due to platelet-specific activation/deletion or simply explained by alterations in megakaryocyte cells which are also targeted by anti-GPIIb antibody. Megakaryocytes were shown to specifically regulate the VWF+ HSC compartment as shown by the Frenette lab (PMID: 29456137). This previous study was not discussed in light of the author's results, and it significantly diminishes the novelty of this study if the effect is mainly derived from megakaryocyte cells.

"It's also missing to evaluate the effect of anti-GPIIb treatment on the levels of PF4, TGFb, Fgf1 and TPO."

This critical control is still not properly addressed. The expression of Pf4, Fgf1 and Tgfb ligands should be measured by ELISA in the BMEF, and not by providing RNAseq data from subsets of selected sub-populations such as EC only isolated in the central marrow. Accordingly, there are no data to support a potential synergistic effect of IL1 with Thpo and CBM-EC-derived PF4 in the regulation of HSC activation post-platelet depletion, as stated in lines 252-256. Most importantly, the contribution of bone marrow EC-derived PF4 compared with megakaryocyte cells or stored in the platelet granules is negligible. This is a critical concern to explain the HSC activation observed, does anti-GPIIb treatment leads to alterations in the BM levels of PF4, TGFb or FGF that could per se drive HSC cell cycle activation, independently of IL1? This is one of the most likely explanations as the results observed are still very small in Il1r1-null mice and LepR-Cre, Il1r1-null mice.

Also, overall, much of the differences in VWF+ and VWF- activation are minor. Although the authors separated the data into main Fig. 1h and Extended Data Fig. 2J, the fact that the biotin MFI of VWF+ and VWF- HSCs is at the exact same level two days post-platelet depletion suggests that both populations have undergone a similar amount of proliferation during this short time.

Finally, it is very unsuitable that the authors instead of clearly addressing the issues raised by this reviewer decided to reply to three of this reviewer's concerns with quotes from a different reviewer (Rev.# 2) highlighting his favorable opinion of the manuscript. These quotes did not provide any clarification of the issues in question, so it's unclear why the authors felt that was an appropriate answer.

Minor comments:

It would be interesting to explain in the context of anti-GPIIb treatment the relationship between cycling VWF+ HSCs and SL-MkPs (stem-like megakaryocyte-committed progenitors, also in the phenotypic HSC compartment) which were shown to undergo rapid and efficient cell cycle entry (16 hr post pl:C treatment) to replenish platelets lost during inflammatory insult, as shown the Essers lab (PMID: 26299573).

Line 166, is this Fig 1h?, there's no extended data Fig 1h

Fig 3i- Cell cycle analysis of VWF+ HSCs legend mentions several controls were used Il1r1FL/+ Vav-CreTg/+, Il1r1+/+ Vav-CreTg/+ and Vav-Cre+/+ mice, however, the figure does not show all these data.

Luis et al: Perivascular niche cells sense thrombocytopenia and activate hematopoietic stem cells in an IL-1 dependent manner (NCOMMS-22-12571A-Z Revised)

RESPONSES TO REVIEWER COMMENTS (All revisions are underlined in the revised manuscript)

Reviewer #1 :

The revision addressed many of the initial concerns. However, the main concerns still remain. The conceptual advance is limited given what is known.

Response: We thank this reviewer for his/her new comments and for acknowledging that the revised manuscript had *“addressed many of the initial concerns”*. We also appreciate the two remaining specific points (Points 2 and 3 below) raised by this reviewer, which we have addressed as outlined in our point-by-point responses below, thereby helping to improve the presentation of the data and incorporation of what we agree are important control data (all revisions are underlined in the revised manuscript). As for the remaining major concern raised by this reviewer regarding what s/he feels is the limited conceptual advance of the manuscript (Point 1), this is obviously as always a matter of judgement, and in that regard we appreciate that the editor stated in his decision letter that *“with regards to the comments of reviewer #1 regarding ‘conceptual advance’, we are satisfied editorially in this regard and consider the conceptual advance your manuscript represents suitable for Nature Communications.”*. Never-the-less, as outlined in more detail below (Point 1) we have in the further revised manuscript tried to better acknowledge the previous important contributions mentioned by this reviewer while at the same time highlighting the limitations in the interpretation of the findings of those studies compared to those in our studies.

1. Regarding the conceptual novelty of the manuscript, I appreciate that the current manuscript used rigorous mouse genetics to conditionally delete IL1R from bone marrow niche cells. However, both Herd et al and Ramasz et al provided evidence that platelet depletion can regulate HSCs by regulating the bone marrow niche. Herd et al specifically focused on LepR+ cells. Ramasz et al demonstrate that the HSC response to acute thrombocytopenia is mediated by SCF and Vegf from the niche. In addition, the phenotypes described by the authors in the IL1R conditional knockout mice are mild (not sure if the mice have a strong platelet production defect after depletion), suggesting this may not be the major pathway mediating HSC activation. Considering all of these points, I am not convinced that the current manuscript represents a significant conceptual advance.

Response: We appreciate that this reviewer specifically points out that our studies used *“rigorous mouse genetics to conditionally delete IL1R from bone marrow niche cells”* as we believe that those genetic experiments were crucial to allow us to demonstrate the role of IL-1 signaling through LepR+ cells in regulating HSCs in response to antibody-induced platelet activation and depletion, distinguishing our studies from those referred to by the reviewer (Herd et al. 2021; Ramasz et al. 2019). The referenced studies (Herd et al. 2021; Ramasz et al. 2019) should indeed be acknowledged for demonstrating that HSCs can be activated by platelet depletion and for providing findings compatible with the role of specific regulators potentially expressed in HSC niches in the bone marrow (something we have tried to do more clearly in the revised manuscript (Page 3, Line 100; Page 5, Line 192). However, in those studies critical genetic experiments to demonstrate the role of a specific niche cell types or the implicated signalling pathways were either not performed (Herd et al. 2021) or, were performed by administration of inhibitors which act systemically to inactivate the signalling pathways in all cells of the body, or by depleting megakaryocytes instead of knocking out the specific signalling pathways in these cells (Ramasz et al. 2019). We tried to clarify this in a more

balanced manner in the revised introduction and results of the manuscript (Page 3, Line 100; Page 5, Line 192). Moreover, our studies link the process of niche-mediated HSC response specifically to platelets activation. We agree that *“the phenotypes described” ... “in the Il1R conditional knockout mice are mild”, “suggesting this may not be the major pathway mediating HSC activation”,* something we also point out in the discussion (Page 11, Line 438), emphasizing that also additional mechanisms are likely to be involved. In fact, it would be surprising if the observed emerging activation of HSCs would be mediated by a single pathway/mechanism. Rather, it is the involvement of a specific niche cell type (LepR+ perivascular cells) and a specific signaling pathway (IL-1R) in the activation of HSCs that represents the main novel finding in our manuscript.

2. The manuscript still needs more careful editing as there are still typos/incorrect citations that make the manuscript hard to understand. For example, Line 166, Ex Data Fig 1h showing H2B-cherry data do not exist. Line 241, it is not clear why IL1a expression by MKs are in line with reports that IL1 is mainly produced by T cells and granulocytes. Line 318, it should be 4g, not 5g.

Response: We thank the reviewer for highlighting these points. We have corrected the specific errors identified by the reviewer and have also carefully double-checked the revised manuscript to ensure that any additional errors could be addressed. The H2B-cherry data, incorrectly referenced as shown in Extended Data Fig.1h are actually shown in Extended Data Fig.2h (revised page 5, Line 164); Regarding the statement about IL-1 expression in bone marrow, the main production of IL-1 by T cells and granulocytes is in line with almost undetectable expression of IL-1 in most bone marrow niche cells, although it is also expressed in Mks, something we show in Extended data Fig.4b. This has been clarified on Page 6, line 239 of the revised manuscript. Finally, it is correct that our referencing to Fig.5g should have been Fig.4g and this has now been corrected (revised Page 9, Line 324).

3. Data in several places need appropriate controls or more information to support the conclusions. For example, IgG control is needed for Extended Data Fig 1e. T cell reconstitution is needed for Fig. 1l and Extended Data Fig 2i. Control Wt + IgG is needed for Fig. 3e. How were Mks purified for RNA-seq in Extended Data Fig 4b? Mks are technically challenging to sort using flow cytometry.

Response: We thank the reviewer for pointing out the importance of including additional control data to better support some of the conclusions in the manuscript. We have now included all the specific controls requested as listed below. Importantly, these do not affect any of the conclusions drawn from the relevant data.

- A representative image of Mk quantification in IgG-treated mice is provided in revised Extended Data Fig.1e and the combined data of all images from IgG and anti-GPIIb treated mice are shown in Extended Data Fig 1g.

- T-cell reconstitution data has as requested been included in Fig1i and Extended data Fig.2i. Importantly, this data corroborates the multilineage reconstitution of both proliferative and non-proliferative HSC subsets.

- Data from the control Wt+IgG group has been added to Fig.3e, showing no differences in HSC cell cycle status when comparing Wt and Il1r1 deficient mice in homeostasis (IgG-treated). This is in line with previous studies reporting a normal HSC compartment in Il1r1^{-/-} mice (Pietras et al. 2016).

- We agree that it is technically challenging to analyse and sort Mks using flow cytometry, and have therefore provided in the revised manuscript new data to better document this. Mks were defined as FSC^{Hi}SSC^{Hi}CD150⁺Vwf-GFP⁺CD41⁺. The FACS sorting gating strategy for the purification of primary mouse MKs according to this phenotypic definition has been included in revised Extended Data Fig.4c (referenced in Page 7, Line 242). To further validate and document the Mk identity of the sorted cells, we have also included in the revised manuscript gene expression (RNA-sequencing) data performed on the sorted cells, showing high expression of known Mk-associated genes as well as undetectable/very low expression of genes characteristic of other blood cell lineages (revised

Extended Data Fig.4d), and thus confirming the correct isolation of MKs. A description of the Mk isolation procedure has also been added to the methods section of the revised manuscript (page 18; Line 777 of the revised manuscript). Briefly, a 2.0 neutral density filter was used on the cell sorter to decrease FSC signal and allow visualization of large cells, and a 100um nozzle was used to preserve viability of the sorted Mk cells.

Reviewer #2:

I was supportive of this manuscript previously and I feel it has only been strengthened by the changes the authors have made. An interesting and robustly-conducted study.

We thank this reviewer for highlighting their support of the original version of the manuscript and acknowledging it has been further strengthened by the revisions.

Reviewer #3:

In response to the reviewer's comments, the authors modified the manuscript and addressed some of the original concerns.

Response: We thank the reviewer for acknowledging that we have addressed some of the initial concerns. With regard to the remaining concerns we have addressed these as outlined in our point-by-point responses below. All revisions are underlined in the revised manuscript.

1) However, one of the key issues it's still not addressed. It's unclear if the observed effects are in fact due to platelet-specific activation/deletion or simply explained by alterations in megakaryocyte cells which are also targeted by anti-GPIIb antibody. Megakaryocytes were shown to specifically regulate the VWF+ HSC compartment as shown by the Frenette lab (PMID: 29456137). This previous study was not discussed in light of the author's results, and it significantly diminishes the novelty of this study if the effect is mainly derived from megakaryocyte cells.

Response: We agree that the important point about the relative role of platelets and megakaryocytes in regulating HSCs in the bone marrow deserves to be addressed in further detail, also because we believe our findings provide some insights into this topic. As the reviewer correctly states this issue is complicated by the fact that anti-GPIIb antibody can target Mks in addition to platelets. Likewise, while we agree that the mentioned studies by the Frenette Lab (Pinho et al, Dev Cell, 2018; reference 33 in revised manuscript; Bruns et al, Nat. Med., 2014) and others (Zhao et al, Nat. Med., 2014) have implicated a role of Mks in the regulation of HSCs, the genetic experiments performed to demonstrate this would also have affected platelets, by including targets shared by both Mks and platelets. The results from the experiments described in Fig 6h in which the mice treated with the anti-GPIIb antibody were previously platelet-depleted by injection of Neuraminidase (Fig 6b) that does not affect MKs (Bruns et al. 2014; Stenberg et al. 1991), provide some important new insights. In this setting injecting anti-GPIIb antibody into NEU-treated mice resulted in reduced HSC activation, when compared to injecting antibody into mice with intact platelet numbers (non-NEU treated), establishing the role of platelets in the observed activation of HSCs (Page 9, Line 357 of the revised text). However, the fact that some HSC activation was observed in mice with NEU-depleted platelets when compared to mice with normal platelet numbers (Fig 1d and Fig. 3e), also supports a role of Mks in the observed HSC activation in response to the to anti-GPII α antibody treatment (Page 10, Line 364 of the revised text). We have also revised Figure 6j and the discussion on page 12, Line 471 to better highlight this issue.

2) "It's also missing to evaluate the effect of anti-GPIIb treatment on the levels of PF4, TGFb, Fgf1 and TPO." This critical control is still not properly addressed. The expression of Pf4, Fgf1 and Tgfb ligands should be measured by ELISA in the BMEF, and not by providing RNAseq data from subsets of selected sub-populations such as EC only isolated in the central marrow. Accordingly, there are no data to support a potential synergistic effect of IL1 with Thpo and CBM-EC-derived PF4 in the regulation of HSC activation post-platelet depletion, as stated in lines 252-256. Most importantly, the contribution of bone marrow EC-derived PF4 compared with megakaryocyte cells or stored in the platelet granules is negligible. This is a critical concern to explain the HSC activation observed, does anti-GPIIb treatment leads to alterations in the BM levels of PF4, TGFb or FGF that could per se drive HSC cell cycle activation, independently of IL1? This is one of the most likely explanations as the results observed are still very small in Il1r1-null mice and LepR-Cre, Il1r1-null mice.

Response: We have now followed up this reviewer's specific request that "*the expression of Pf4, Fgf1 and Tgfb ligands should be measured by ELISA in the BMEF*", factors previously implicated in the regulation of HSCs by MKs (Bruns et al. 2014; Zhao et al. 2014). We measured this in the bone marrow extracellular fluid isolated from mice in homeostasis (IgG treated) and post platelet depletion (GPIIb-treated). While no differences were observed in the levels of TGFb1 and FGF1 a significant increase was observed in the level of PF4 (Extended Data Fig.4g; Page 7, Line 258 of the revised manuscript). Interestingly, Mk-derived PF4 has been previously demonstrated to induce HSC quiescence (Bruns et al, 2014) and in line with this, PF4 administration to mice leads to reduced numbers of Vwf+ HSCs (Pinho et al, 2014). Therefore, the observed increased levels of PF4 is unlikely to explain the activation of HSC proliferation we report here. Rather, it raises the possibility that the increased levels of PF4 represents a negative feedback loop to bring HSCs back to quiescence, which happens quite quickly after the initial activation of HSC proliferation (Fig.1c-d). While we agree that this analysis of protein levels of these factors in the bone marrow is more informative with regard to a potential mechanistic role, the RNA sequencing analysis included in the manuscript provides information on the cellular source of these factors, which cannot be obtained from the ELISA measurements. We initially provided this data for Mks, previously shown to express these factors (Bruns et al. 2014; Zhao et al. 2014), and for CBM populations, which are the cells where we observed significant transcriptional changes post platelet depletion. Following this reviewer's point, we have now included the gene expression analysis on all cell populations investigated in this study (Extended Data Fig.4f; Page 7, Line 256). While expression of *Pf4* was only observed in Mks and CBM-ECs, the expression of *Tgfb1* and *Fgf1* is not restricted to MKs and is broadly detected across different niche cell populations. Of note and in line with the ELISA measurements, GPIIb-mediated platelet depletion leads to a significant increase in the expression of *Pf4* in CBM-ECs. While our data shows, as indicated by this reviewer, that Mks express *Pf4* at much higher levels than CBM-ECs, ECs are much more abundant in BM and an integrated part of HSC niches. Therefore, the >20-fold upregulation of this gene in CBM-ECs raises the interesting possibility that in addition to Mks, ECs may constitute a relevant source of PF4 in situations of perturbed hematopoiesis, possibly to restore HSC quiescence. The expression of *Pf4* in other niche cells including Mks, and of *Tgfb1* and *Fgf1* in all niche cells analysed was not significantly altered following platelet depletion.

Thus, both the ELISA measurements of TGFb1, PF4 and FGF1 in the BM extracellular fluid and the RNA sequencing analysis in different niche cell populations, did not provide evidence that Mk-derived TGFb1, PF4 and FGF1 regulate the activation of HSC proliferation post GPIIb-mediated platelet depletion. Nevertheless, analysis of total BM fluid cannot detect very localized changes in the concentration of these factors, which might be relevant for the HSCs activation. Definitive evidence of the potential involvement of these (and other) factors in the activation (and/or return to quiescence) of HSCs post platelet depletion will require further genetic experiments, which we hope the reviewer agrees are beyond the scope of this study. We discuss this in the revised manuscript (Page 7, Line 251; Page 11, Line 438).

3) Also, overall, much of the differences in VWF+ and VWF- activation are minor. Although the authors separated the data into main Fig. 1h and Extended Data Fig. 2J, the fact that the biotin MFI of VWF+ and VWF- HSCs is at the exact same level two days post-platelet depletion suggests that both populations have undergone a similar amount of proliferation during this short time.

Response: Cell cycle analysis of HSCs 1 day post platelet depletion showed a preferential recruitment of Vwf+ HSCs into active proliferation (S-G2-M) (Fig.1c-d). While the cell cycle analysis provides a snapshot of the cell cycle distribution stages of a population at specific time points, the methods applied here to investigate proliferation (dilution of Biotin and H2B-mCherry labelling) provide an history of proliferation, which is also transferred to their progeny when cells divide and differentiate. This might be relevant for the lack of a clear difference in the proliferation history of Vwf+ and Vwf- HSCs, in light of the hierarchical relationship we have previously established to exist between Vwf+ and Vwf- HSCs, with Vwf+ HSCs giving rise to Vwf- HSCs and not vice versa (Sanjuan-Pla et al. 2013). Regardless the consequence of the platelet activation and depletion is a broad activation of HSCs. We discuss this in the revised manuscript (Page 5, Line 173; Page 10, Line 397). Realizing from the outset this limitation of the biotin dilution assay, it is important to emphasize (as it is on Page 4, Line 157; Page 5, Line 184; Page 10, Line 393) that the primary purpose of this assay was to unequivocally establish that the phenotypically defined HSCs included functionally defined LT-HSCs.

4) Finally, it is very unsuitable that the authors instead of clearly addressing the issues raised by this reviewer decided to reply to three of this reviewer's concerns with quotes from a different reviewer (Rev.# 2) highlighting his favorable opinion of the manuscript. These quotes did not provide any clarification of the issues in question, so it's unclear why the authors felt that was an appropriate answer.

Response: In hindsight we can only agree (and apologize) that our approach was not very constructive, although we obviously also tried to specifically address the specific points raised.

Minor comments:

5) It would be interesting to explain in the context of anti-GPIIb treatment the relationship between cycling VWF+ HSCs and SL-MkPs (stem-like megakaryocyte-committed progenitors, also in the phenotypic HSC compartment) which were shown to undergo rapid and efficient cell cycle entry (16 hr post pl:C treatment) to replenish platelets lost during inflammatory insult, as shown the Essers lab (PMID: 26299573).

Response: While we did not specifically investigate this, in addition to HSCs, our findings are also compatible with the involvement of committed Mk progenitors such as stem-like Mk-committed progenitors (Haas et al. 2015), in the observed rapid response to platelet depletion and activation. We have now highlighted this in the revised results (Page 4, Line 156) and discussion (Page 13, Line 489). However, while sharing phenotypic similarities with true HSCs, the authors of that study demonstrated themselves that SL-MkPs do not have long-term reconstitution capacity (Haas et al. 2015), whereas we demonstrate the activation of functionally defined HSCs that proliferate while retaining multilineage reconstitution capacity, as shown by transplantation of Biotin and H2B-mCherry labelled HSCs (Fig.1h-j; Ext Data Fig.2h-m).

6) Line 166, is this Fig 1h?, there's no extended data Fig 1h

Response: We apologise for this error. The data referenced are presented in Extended data Fig.2h, and this is corrected in the revised manuscript.

7) Fig 3i- Cell cycle analysis of VWF+ HSCs legend mentions several controls were used $Il1r1^{FL/+}$ Vav-CreTg/+, $Il1r1^{+/+}$ Vav-CreTg/+ and Vav-Cre+/+ mice, however, the figure does not show all these data.

Response: In the experiments with Vav-Cre^{Tg/+} $Il1r1^{FL/FL}$ mice we used as controls a mixture of relevant genotypes obtained through the required breeding, as indicated in the legend of Figure 3i. None of these genotypes result in a IL-1R loss of function, as now stated in the revised legend of Figure 3. To demonstrate that these genotypes gave comparable results we have below plotted the same graph as in Fig.3i but with the different genotypes used as controls in different colours. For simplicity and to not distract from the main finding in the data shown (which is anyhow a lack of phenotype) we have opted for only showing this graph here without including it in the revised manuscript. We are however happy to include this in a supplementary figure if this reviewer or the editor deems it necessary.

Fig.1 Cell cycle analysis of Vwf -tdTomato⁺ HSCs from mice with conditional deletion of $Il1r1$ in all hematopoietic cells ($Il1r1^{FL/FL}$ Vav-Cre^{Tg/+}) 1 day post platelet depletion. Controls include Vwf -tdTomato⁺ HSCs from $Il1r1^{FL/+}$ Vav-Cre^{Tg/+}, $Il1r1^{+/+}$ Vav-Cre^{Tg/+} and Vav-Cre^{+/+} mice. Data represent mean \pm SD frequencies of 4-5 mice per condition from 3 independent experiments. ***, $p < 0.001$; * $p < 0.05$; ns, non-significant ($p > 0.05$) using 2-way ANOVA with Tukey's multiple comparisons.

References

- Bruns, I., D. Lucas, S. Pinho, J. Ahmed, M. P. Lambert, Y. Kunisaki, C. Scheiermann, L. Schiff, M. Poncz, A. Bergman, and P. S. Frenette. 2014. 'Megakaryocytes regulate hematopoietic stem cell quiescence through CXCL4 secretion', *Nat Med*, 20: 1315-20.
- Haas, S., J. Hansson, D. Klimmeck, D. Loeffler, L. Velten, H. Uckelmann, S. Wurzer, A. M. Prendergast, A. Schnell, K. Hexel, R. Santarella-Mellwig, S. Blaszkiewicz, A. Kuck, H. Geiger, M. D. Milsom, L. M. Steinmetz, T. Schroeder, A. Trumpp, J. Krijgsvelde, and M. A. Essers. 2015. 'Inflammation-Induced Emergency Megakaryopoiesis Driven by Hematopoietic Stem Cell-like Megakaryocyte Progenitors', *Cell Stem Cell*, 17: 422-34.
- Herd, O. J., G. F. Rani, J. P. Hewitson, K. Hogg, A. P. Stone, N. Cooper, D. G. Kent, P. G. Genever, and I. S. Hitchcock. 2021. 'Bone marrow remodeling supports hematopoiesis in response to immune thrombocytopenia progression in mice', *Blood Adv*, 5: 4877-89.
- Nakamura-Ishizu, A., K. Takubo, M. Fujioka, and T. Suda. 2014. 'Megakaryocytes are essential for HSC quiescence through the production of thrombopoietin', *Biochem Biophys Res Commun*, 454: 353-7.
- Pietras, E. M., C. Mirantes-Barbeito, S. Fong, D. Loeffler, L. V. Kovtonyuk, S. Zhang, R. Lakshminarasimhan, C. P. Chin, J. M. Techner, B. Will, C. Nerlov, U. Steidl, M. G. Manz, T. Schroeder, and E. Passegue. 2016. 'Chronic interleukin-1 exposure drives haematopoietic stem cells towards precocious myeloid differentiation at the expense of self-renewal', *Nat Cell Biol*, 18: 607-18.

- Pinho, S., T. Marchand, E. Yang, Q. Wei, C. Nerlov, and P. S. Frenette. 2018. 'Lineage-Biased Hematopoietic Stem Cells Are Regulated by Distinct Niches', *Dev Cell*, 44: 634-41 e4.
- Qian, H., N. Buza-Vidas, C. D. Hyland, C. T. Jensen, J. Antonchuk, R. Mansson, L. A. Thoren, M. Ekblom, W. S. Alexander, and S. E. Jacobsen. 2007. 'Critical role of thrombopoietin in maintaining adult quiescent hematopoietic stem cells', *Cell Stem Cell*, 1: 671-84.
- Ramasz, B., A. Kruger, J. Reinhardt, A. Sinha, M. Gerlach, A. Gerbaulet, S. Reinhardt, A. Dahl, T. Chavakis, B. Wielockx, and T. Grinenko. 2019. 'Hematopoietic stem cell response to acute thrombocytopenia requires signaling through distinct receptor tyrosine kinases', *Blood*, 134: 1046-58.
- Sanjuan-Pla, A., I. C. Macaulay, C. T. Jensen, P. S. Woll, T. C. Luis, A. Mead, S. Moore, C. Carella, S. Matsuoka, T. Bouriez Jones, O. Chowdhury, L. Stenson, M. Lutteropp, J. C. Green, R. Facchini, H. Boukarabila, A. Grover, A. Gambardella, S. Thongjuea, J. Carrelha, P. Tarrant, D. Atkinson, S. A. Clark, C. Nerlov, and S. E. Jacobsen. 2013. 'Platelet-biased stem cells reside at the apex of the haematopoietic stem-cell hierarchy', *Nature*, 502: 232-6.
- Stenberg, P. E., J. Levin, G. Baker, Y. Mok, and L. Corash. 1991. 'Neuraminidase-induced thrombocytopenia in mice: effects on thrombopoiesis', *J Cell Physiol*, 147: 7-16.
- Yamamoto, R., Y. Morita, J. Ooehara, S. Hamanaka, M. Onodera, K. L. Rudolph, H. Ema, and H. Nakauchi. 2013. 'Clonal analysis unveils self-renewing lineage-restricted progenitors generated directly from hematopoietic stem cells', *Cell*, 154: 1112-26.
- Yoshihara, H., F. Arai, K. Hosokawa, T. Hagiwara, K. Takubo, Y. Nakamura, Y. Gomei, H. Iwasaki, S. Matsuoka, K. Miyamoto, H. Miyazaki, T. Takahashi, and T. Suda. 2007. 'Thrombopoietin/MPL signaling regulates hematopoietic stem cell quiescence and interaction with the osteoblastic niche', *Cell Stem Cell*, 1: 685-97.
- Zhao, M., J. M. Perry, H. Marshall, A. Venkatraman, P. Qian, X. C. He, J. Ahamed, and L. Li. 2014. 'Megakaryocytes maintain homeostatic quiescence and promote post-injury regeneration of hematopoietic stem cells', *Nat Med*, 20: 1321-6.

REVIEWERS' COMMENTS

Reviewer #1 (Remarks to the Author):

The revision has largely addressed my technical concerns. There is a suggestion for the authors to edit the manuscript. In response #2, the part on IL1a expression by MKs still reads oddly. Steady-state IL-1 expression by T cells and granulocytes does not infer whether MKs express IL1a. If the authors want to identify the source of IL1a, they should look at T cells and granulocytes as well.

As stated by the authors, the involvement of IL-1R pathway in niche cells (as shown with mouse genetics compared with previous studies) is the main novelty. However, the phenotypes of the genetic models are really mild (an at-most 30% reduction of platelet count in peripheral blood in Fig. 5c). This undermines the importance of the proposed mechanisms and the impact of the work. The authors added discussions on potential involvement of other factors/pathways, which helps to some extent. But no evidence suggests that these other factors act through the niche.

Reviewer #3 (Remarks to the Author):

The authors addressed all my concerns, and the revised manuscript improved significantly.

Response to reviewers

REVIEWERS' COMMENTS

Reviewer #1 (Remarks to the Author):

1) The revision has largely addressed my technical concerns. There is a suggestion for the authors to edit the manuscript. In response #2, the part on IL1a expression by MKs still reads oddly. Steady-state IL-1 expression by T cells and granulocytes does not infer whether MKs express IL1a. If the authors want to identify the source of IL1a, they should look at T cells and granulocytes as well.

Response: We thank the reviewer for stating that the revision has largely addressed his/her technical concerns. We have further revised the relevant statement (page 6, Line 238-242) not to infer anything about IL-1 expression in megakaryocytes based on previously established expression in other cell types: "In steady-state IL-1 has been shown to mainly be produced by circulating T cells (IL-1 α) and granulocytes (IL-1 β). In our datasets analyzing for IL-1 expression both Il1a and Il1b were mostly undetectable in the different niche cell populations as well as in HSCs (Supplementary Fig.4b), whereas primary Mks (Supplementary Fig.4c-d) showed high expression, in particular of Il1a (Supplementary Fig.4b)."

2) As stated by the authors, the involvement of IL-1R pathway in niche cells (as shown with mouse genetics compared with previous studies) is the main novelty. However, the phenotypes of the genetic models are really mild (an at-most 30% reduction of platelet count in peripheral blood in Fig. 5c). This undermines the importance of the proposed mechanisms and the impact of the work. The authors added discussions on potential involvement of other factors/pathways, which helps to some extent. But no evidence suggests that these other factors act through the niche.

Response: We agree that other factors must be involved in the activation of HSCs after platelet depletion/activation, and have further revised the paragraph of the discussion in which we discuss this to emphasize that it also remains unclear whether these additional factors would act through the niche (page 12, Line 454): "Regardless, further genetic studies will be required to establish the role of additional regulators in the IL-1-independent HSC proliferation following platelet activation and depletion, and whether they act through cells in HSC niches".

Reviewer #3 (Remarks to the Author):

The authors addressed all my concerns, and the revised manuscript improved significantly.

Response: We thank the reviewer for the positive feedback on our manuscript.